# Divide, Evaluate, and Refine: Evaluating and Improving Text-to-Image Alignment with Iterative VQA Feedback

**Jaskirat Singh**[1]
[1]The Australian National University

**Liang Zheng**[1,2]
[2]Australian Centre for Robotic Vision

https://1jsingh.github.io/divide-evaluate-and-refine

## Abstract

The field of text-conditioned image generation has made unparalleled progress with the recent advent of latent diffusion models. While remarkable, as the complexity of given text input increases, the state-of-the-art diffusion models may still fail in generating images which accurately convey the semantics of the given prompt. Furthermore, it has been observed that such misalignments are often left undetected by pretrained multi-modal models such as *CLIP*. To address these problems, in this paper we explore a simple yet effective decompositional approach towards both evaluation and improvement of text-to-image alignment. In particular, we first introduce a ***Decompositional-Alignment-Score*** which given a complex prompt decomposes it into a set of disjoint assertions. The alignment of each assertion with generated images is then measured using a VQA model. Finally, alignment scores for different assertions are combined aposteriori to give the final text-to-image alignment score. Experimental analysis reveals that the proposed alignment metric shows significantly higher correlation with human ratings as opposed to traditional *CLIP, BLIP* scores. Furthermore, we also find that the assertion level alignment scores provide a useful feedback which can then be used in a simple iterative procedure to gradually increase the expression of different assertions in the final image outputs. Human user studies indicate that the proposed approach surpasses previous state-of-the-art by 8.7% in overall text-to-image alignment accuracy.

## 1  Introduction

The field of text-to-image generation has made significant advancements with the recent advent of large-scale language-image (LLI) models [1–5]. In particular, text-conditioned latent diffusion models have shown unparalleled success in generating creative imagery corresponding to a diverse range of free-form textual descriptions. However, while remarkable, it has been observed [6–8] that as the complexity of the input text increases, the generated images do not always accurately align with the semantic meaning of the textual prompt.

To facilitate the reliable use of current text-to-image generation models for practical applications, it is essential to answer two key questions: 1) Can we detect such fine-grain misalignments between the input text and the generated output in a robust manner? and 2) Once detected, can we improve the text-to-image alignment for failure cases? While several metrics for evaluating text-to-image alignment (*e.g.*, CLIP [9], BLIP [10], BLIP2 [11]) exist, it has been observed [7, 12] that a high score with these metrics can be achieved even if the image does not fully correspond with input prompt. For instance, in Fig. 1, an output image (containing only pink trees) shows high CLIP/BLIP scores with the text "pink trees and yellow car" even if yellow car is not present. Evaluating text-to-image matching using the image-text-matching (ITM) head of BLIP models has also been recently explored [10, 11]. However, the generated scores also show a similar tendency to favor the main subject

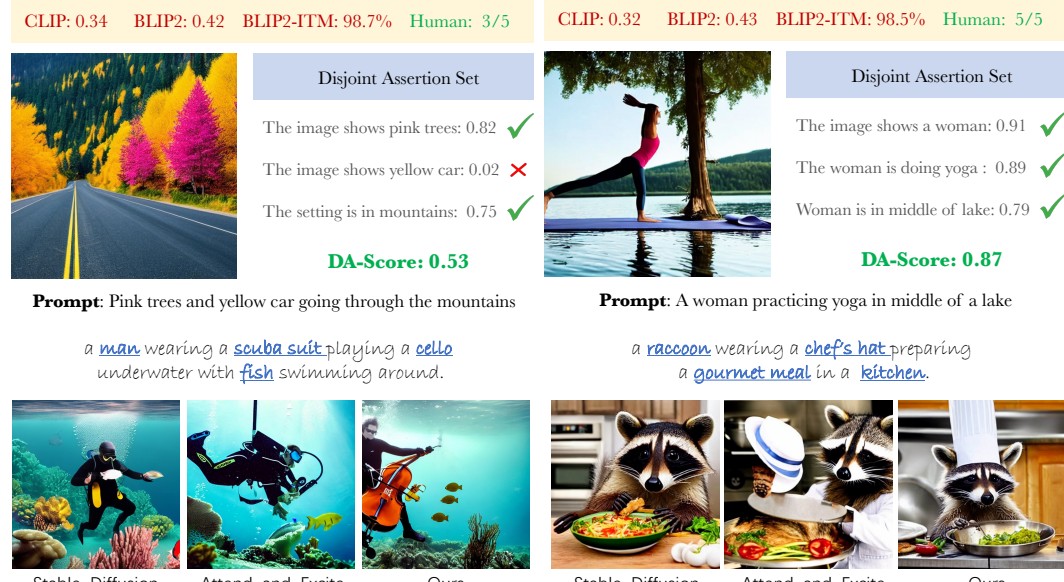

Figure 1: *Overview. Top:* Traditional methods for evaluating text-to-image alignment *e.g.*, CLIP [9], BLIP-2 [10] and BLIP2-ITM (which provides a binary image-text matching score between 0 and 1) often fail to distinguish between good (*right*) and bad (*left*) image outputs and can give high scores even if the generated image is not an accurate match for input prompt (missing yellow car). In contrast, by breaking down the prompt into a set of disjoint assertions and then evaluating their alignment with the generated image using a VQA model [10], the proposed Decompositional-Alignment Score (DA-score) shows much better correlation with human ratings (refer Sec. 4.1). *Bottom:* Furthermore, we show that the assertion-level alignment scores can be used along with a simple iterative refinement strategy to reliably improve the alignment of generated image outputs (refer Sec. 4.2).

of input prompt. Furthermore, even if such misalignments are detected, it is not clear how such information can be used for improving the quality of generated image outputs in a reliable manner.

To address these problems, in this paper we explore a simple yet effective decompositional approach towards both evaluation and improvement of fine-grain text-to-image alignment. In particular, we propose a ***Decompositional-Alignment-Score*** (DA-Score) which given a complex text prompt, first decomposes it into a set of disjoint assertions about the content of the prompt. The alignment of each of these assertions with the generated image is then measured using a VQA model [10, 13]. Finally, the alignment scores for different assertions are combined to give an overall text-to-image alignment score. Our experiments reveal that the proposed evaluation score shows significantly higher correlation with human ratings over prior evaluation metrics (*e.g.*, CLIP, BLIP, BLIP2) (Sec. 4.1).

Furthermore, we also find that the assertion-level alignment scores provide a useful and explainable feedback for determining which parts of the input prompt are not being accurately described in the output image. We show that this feedback can then be used to gradually improve the alignment of the generated images with the input text prompt. To this end, we propose a simple iterative refinement procedure (Fig. 2), wherein at each iteration the expressivity of the least-aligned assertion is improved by increasing the weightage/cross-attention strength (refer Sec. 3.2) of corresponding prompt tokens during the reverse diffusion process. Through both qualitative and quantitative analysis, we find that the proposed iterative refinement process allows for generation of better aligned image outputs over prior works [6–8] while on average showing comparable inference times (Sec. 4.2).

## 2   Related Work

**Text to Image Generation Models.** Text conditional image synthesis is a topic of keen interest in the vision community. For instance, [14–18] use GANs to perform text guided image generation. Similarly, [5, 19] explore the use of autoregressive models for zero-shot text to image generation. Recently, diffusion-based-models [1–5, 20, 21] have emerged as a powerful class of methods for performing text-conditional image synthesis over diverse range of target domains.

a couple wearing scuba gear having a tea party underwater with a school of fish

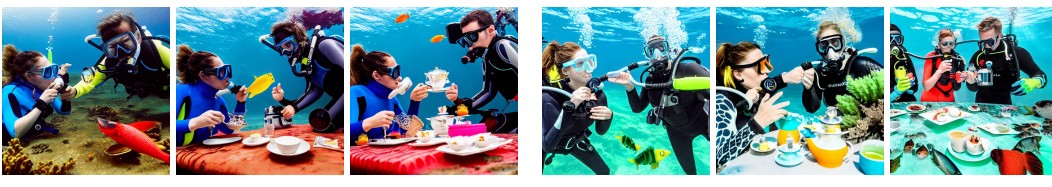

a man riding a skateboard down a mountain road while holding an umbrella and wearing goggles.

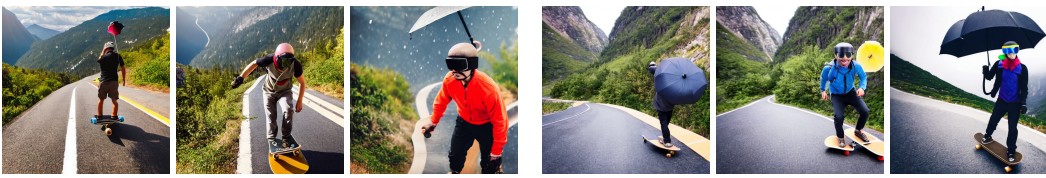

Figure 2: *Iterative refinement (Col:1-3;4-6) for improving text-to-image alignment.* We propose a simple iterative refinement approach which uses the decompositional alignment scores (refer Sec. 3.1) as feedback to gradually improve the alignment of the generated images with the input text-prompt.

While remarkable, generating images which align perfectly with the input text-prompt remains a challenging problem [6–8, 22]. To enforce, heavier reliance of generated outputs on the provided text, classifier-free guidance methods [2, 3, 23] have been proposed. Similarly, use of an additional guidance input to improve controllability of text-to-image generation have recently been extensively explored [24–35]. However, even with their application, the generated images are often observed to exhibit fine-grain misalignments such as missing secondary objects [6, 7] with the input text prompt.

**Evaluating Image-Text Alignment.** Various protocols for evaluating text-image alignment in a reference-free manner have been proposed [9–11]. Most prior works [2, 3, 5, 9] typically use the cosine similarity between the text and image embedding from large-scale multi-modal models [9, 36–38] such as CLIP [9], BLIP [10], BLIP-2 [11] for evaluating the alignment scores. Recently, [10, 11] also show the application of BLIP/BLIP-2 models for image-text matching using image retrieval. However, as shown in Fig. 1, these scores can give very high scores even if the generated images do not full align with the input text prompt. Furthermore, unlike our approach image-text alignment is often represented through a single scalar value which does not provide an explainable measure which can be used to identify/improve weaknesses of the image generation process.

**Improving Image-Text Alignment.** Recently several works [6–8] have been proposed to explore the problem of improving image-text alignment in a training free manner. Liu *et al*. [6] propose to modify the reverse diffusion process by composing denoising vectors for different image components. However, it has been observed [7] that it struggles while generating photorealistic compositions of diverse objects. Feng *et al*. [8] use scene graphs to split the input sentence into several noun phrases and then assign a designed attention map to the output of the cross-attention operation. In another recent work, Chefer *et al*. [7] extend the idea of cross-attention map modification to minimize missing objects but instead do so by modifying the noise latents during the reverse diffusion process. While effective at reducing missing objects, we find that the performance / quality of output images can suffer as the number of subjects in the input prompt increases (refer Sec. 4.2).

Besides training-free methods, recent contemporary work [39, 40] has also explored the possibility of improving image-text alignment using human feedback to finetune existing latent diffusion models. However this often requires the collection of large-scale human evaluation scores and finetuning the diffusion model across a range of diverse data modalities which can be expensive. In contrast, we explore a training free approach for improvement of fine-grain text-to-image alignment.

## 3  Our Method

Given the image generation output $\mathcal{I}$ corresponding to a text prompt $\mathcal{P}$, we wish to develop a mechanism for evaluation and improvement of fine-grain text-to-image alignment. The core idea of our approach is to take a decompositional strategy for both these tasks. To this end, we first generate a set of disjoint assertions regarding the content of the input prompt. The alignment of the output image $\mathcal{I}$ with each of these assertions is then calculated using a VQA model. Finally, we use the

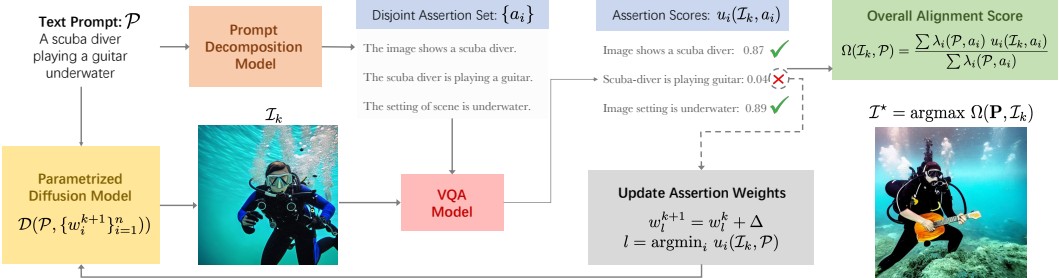

Figure 3: *Method Overview.* Given a text prompt $\mathcal{P}$ and an initially generated output $\mathcal{I}_0$, we first generate a set of disjoint assertions $a_i$ regarding the content of the caption. The alignment of the output image $\mathcal{I}_0$ with each of these assertions is then calculated using a VQA model. Finally, we use the assertion-based-alignment scores $u_i(\mathcal{I}_0, \mathcal{P})$ as feedback to increase the weightage $w_i$ (of the assertion with least alignment score) in a parameterized diffusion model formulation $\mathcal{D}$ (Sec. 3.2). This process can then be performed in an iterative manner to gradually improve the quality of the generated outputs until a desirable threshold for the overall alignment score $\Omega(\mathcal{I}_k, \mathcal{P})$ is reached.

assertion-based-alignment scores as feedback to improve the expressiveness of the assertion with the least alignment score. This process can then be performed in an iterative manner to gradually improve the quality of generated outputs until a desired value for the overall alignment score is attained.

In the next sections, we discuss each of these steps in detail. In Sec. 3.1 we first discuss the process for evaluating decompositional-alignment scores. We then discuss the iterative refinement process for improving text-to-image alignment in Sec. 3.2. Fig. 3 provides an overview for the overall approach.

## 3.1 Evaluating Text-to-Image Alignment

**Prompt Decomposition Model.** Given an input prompt $\mathcal{P}$, we first decompose its textual information into a set of disjoint assertions (and corresponding questions) which exhaustively cover the contents of the input prompt. Instead of relying on human-inputs as in [6, 7][1], we leverage the in-context learning capability [41] of large-language models [42, 43] for predicting such decompositions in an autonomous manner. In particular, given an input prompt $\mathcal{P}$ and large-language model $\mathcal{M}$, the prompt decomposition is performed using in-context learning as,

$$\mathbf{x} = \{x_0, x_1, \ldots x_n\} = \mathcal{M}(\mathbf{x} \mid \mathcal{P}, D_{exemplar}, \mathcal{T}), \tag{1}$$

where $\mathbf{x}$ is the model output, $n$ is the number of decompositions, $D_{exemplar}$ is the in-context learning dataset consisting 4-5 human generated examples for prompt decomposition, and $\mathcal{T}$ is task description. Please refer supp. material for further details on exemplar-dataset and task-description design.

The model output $\mathbf{x}$ is predicted to contain tuples $x_i = \{a_i, p_i\}$, where each tuple is formatted to contain assertions $a_i$ and the sub-part $p_i$ of the original prompt $\mathcal{P}$ corresponding to the generated assertion. For instance, given $\mathcal{P}$ : *'a cat and a dog'* the prompt decomposition can be written as,

$$\mathcal{M}(\mathbf{x} \mid \mathcal{P} : \textit{'a cat and a dog'}, D_{exemplar}, \mathcal{T}) = [\{\textit{'there is a cat'}, \textit{'a cat'}\}, \{\textit{'there is a dog'}, \textit{'a dog'}\}].$$

**Computing Assertion-based Alignment Scores.** We next compute the alignment of the generated image $\mathcal{I}$ with each of the disjoint assertions using a Visual-Question-Answering (VQA) model [10]. In particular, given image $\mathcal{I}$, assertions $a_i, i = 1, \ldots n$, their rephrasing in question format $a_i^q$ and VQA-model $\mathcal{V}$, the assertion-level alignment scores $u_i(\mathcal{I}, a_i)$ are computed as,

$$u_i(\mathcal{I}, a_i) = \frac{\exp\left(\alpha_i/\tau\right)}{\exp\left(\alpha_i/\tau\right) + \exp\left(\beta_i/\tau\right)}, \textit{ where } \quad \alpha_i = \mathcal{V}(\textit{'yes'} \mid \mathcal{I}, a_i^q), \quad \beta_i = \mathcal{V}(\textit{'no'} \mid \mathcal{I}, a_i^q),$$

where $\alpha_i, \beta_i$ refer to the logit-scores of VQA-model $\mathcal{V}$ for input tuple (image $\mathcal{I}$, question $a_i^q$) corresponding to output tokens *'yes'*, *'no'* respectively. Hyperparameter $\tau$ controls the temperature of the softmax operation and controls the confidence of the alignment predictions.

---

[1]Prior works on improving image-text alignment often rely on human-user inputs for expressing contents of the input prompt into its simpler constituents. For instance, Feng *et al*. [6] require the user to describe the prompt as a conjunction/disjunction of simpler statements. Similarly, Chefer *et al*. [7] require the user to provide a set of entities / subjects in the prompt, over which their optimization should be performed.

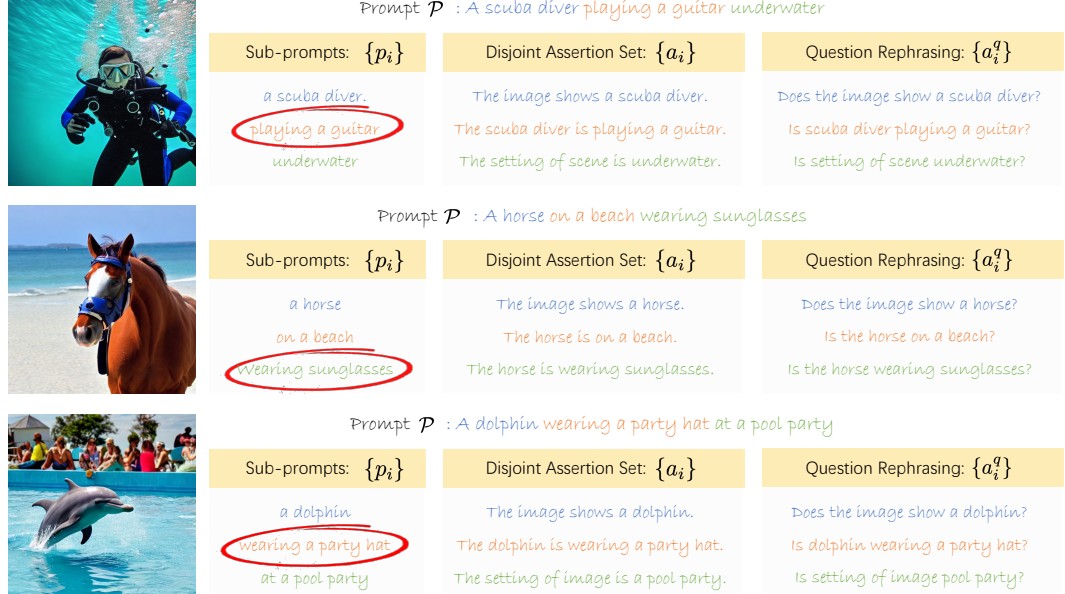

Figure 4: *Visualizing the prompt decomposition process.* By dividing a complex prompt $\mathcal{P}$ into a set of disjoint assertions $a_i$, we are able to identify the sub-prompts $p_i$ (circled) which are not expressed in the image output using VQA, and thereby address them using iterative refinement (Sec. 3.2).

**Combining Alignment Scores.** Finally, the assertion level alignment-scores $u_i(\mathcal{I}, a_i)$ are combined to give the overall text-to-image alignment score $\Omega(\mathcal{I}, \mathcal{P})$ between image $\mathcal{I}$ and prompt $\mathcal{P}$ as,

$$\Omega(\mathcal{I}, \mathcal{P}) = \frac{\sum_i \lambda_i(\mathcal{P}, a_i)\, u_i(\mathcal{I}_k, a_i)}{\sum_i \lambda_i(\mathcal{P}, a_i)}, \tag{2}$$

where weights $\lambda_i(\mathcal{P}, a_i)$ refer to the importance of assertion $a_i$ in capturing the overall content of the input prompt $\mathcal{P}$, and allows the user to control the relative importance of different assertions in generating the final image output[2]. Please refer Fig. 3 for the overall implementation.

## 3.2 Improving Text to Image Alignment

In addition to predicting overall text-to-image alignment score, we find that assertion-level alignment scores $u_i(\mathcal{I}, a_i)$ also provide a useful and explainable way for determining which parts of the input prompt $\mathcal{P}$ are not being accurately described in the output image $\mathcal{I}$. This feedback can then be used in an iterative manner to improve the expressivity of the assertion with least alignment score $u_i(\mathcal{I}, q_i)$, until a desired threshold for the overall text-image alignment score $\Omega(\mathcal{I}, \mathcal{P})$ is obtained.

**Parameterized Diffusion Model**. We first modify the image generation process of standard diffusion models in order to control the expressiveness of different assertions $a_i$ in parametric manner. In particular, we modify the reverse diffusion process to also receive inputs weights $w_i$, where each $w_i$ controls the relative importance of assertion $a_i$ during the image generation process. In this paper, we mainly consider the following two methods for obtaining such parametric control.

**Prompt Weighting.** Instead of computing the CLIP [36] features from original prompt $\mathcal{P}$ we use prompt-weighting [44] to modify the input CLIP embeddings to the diffusion model as,

$$\text{CLIP}(\mathcal{P}) = \mathcal{W}(\mathcal{P}, \{\text{CLIP}(p_i), w_i\}_{i=1}^n)) \tag{3}$$

where $\mathcal{W}$ refers to the prompt-weighting function from [1, 44], $p_i$ refers to the sub-prompt (Sec. 3.1) corresponding to assertion $a_i$, and weights $w_i$ control the relative weight of different sub-prompts $p_i$ in computing the overall CLIP embedding for prompt $\mathcal{P}$.

**Cross-Attention Control.** Similar to [7], we also explore the idea of modifying the noise latents $z_t$ during the reverse diffusion process, to increase the cross-attention strength of the main noun-

---

[2]For simplicity reasons, we mainly use $\lambda_i = 1 \forall i$ in the main paper. Further analysis on variable $\lambda_i$ to account for variable information content or visual verifiability of an assertion are provided in supp. material.

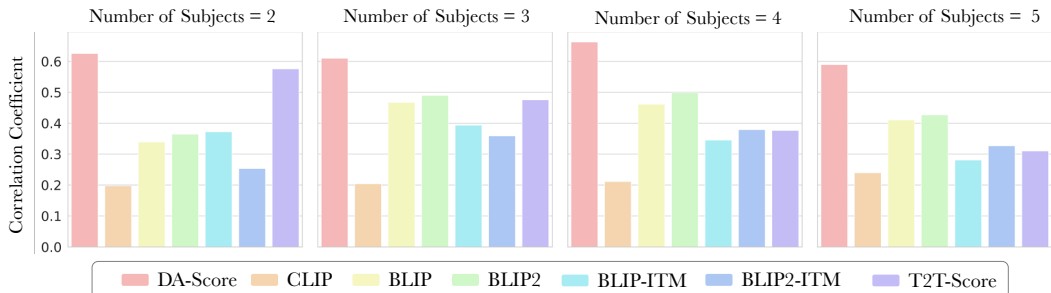

Figure 5: *Method comparisons w.r.t correlation with human ratings.* We compare the correlation of different text-to-image alignment scores with those obtained from human subjects, as the number of subjects in the input prompt (refer Sec. 4) is varied. We observe that the proposed alignment score (DA-score) provides a better match for human-ratings over traditional text-to-image alignment scores.

subject for each sub-assertion $a_i$. However, instead of only applying the gradient update for the least dominant subject [7], we modify the loss for the latent update in parametric form as,

$$z_t = z_t - \alpha \nabla_{z_t} \mathcal{L}(z_t, \{w_i\}_{i=1}^n)), \text{ where } \quad \mathcal{L}(z_t, \{w_i\}_{i=1}^n) = \sum_i w_i(1 - \max G(\mathcal{A}_i^t)), \quad (4)$$

where $\alpha$ is the step-size, $\mathcal{A}_i^t$ refer to the attention map corresponding to the main noun-subject in assertion $a_i$, $G$ is a smoothing function and weights $w_i$ control the extent to which the expression of different noun-subjects in the prompt (for each assertion) will be increased in the next iteration.

**Iterative Refinement.** Given the above parametric formulation for controlling expression of different assertions, we next propose a simple yet effective iterative refinement approach towards improving text-to-image alignment. In particular, at any iteration $k \in [1, 5]$ during the refinement process, we first compute both overall text-image similarity score $\Omega(\mathcal{I}_k, \mathcal{P})$ and assertion-level alignment scores $u_i(\mathcal{I}_k, \mathcal{P})$. The image generation output $\mathcal{I}_{k+1}$ for the next iteration is then computed as,

$$\mathcal{I}_{k+1} = \mathcal{D}(\mathcal{P}, \{w_i^{k+1}\}_{i=1}^n)); \text{ where } \quad w_i^{k+1} = \begin{cases} w_i^k + \Delta, \text{ if } \quad i = \operatorname{argmin}_l u_l(\mathcal{I}, \mathcal{P}) \\ w_i^k \quad \text{otherwise} \end{cases}, \quad (5)$$

where $\mathcal{D}$ refers to the parametrized diffusion model and $\Delta$ is a hyper-parameter. This iterative process is then performed until a desirable threshold for the overall alignment score $\Omega(\mathcal{I}_k, \mathcal{P})$ is reached. The image generation output $\mathcal{I}^\star$ at the end of the refinement process is then computed as,

$$\mathcal{I}^\star = \operatorname{argmax}_{\mathcal{I}_k} \Omega(\mathcal{I}_k, \mathcal{P}). \quad (6)$$

## 4 Experiments

**Dataset.** Since there are no openly available datasets addressing semantic challenges in text-based image generation with human annotations, we introduce a new benchmark dataset Decomposable-Captions-4k for method comparison. The dataset consists an overall of 24960 human annotations on images generated using all methods [1, 6, 7] (including ours) across a diverse set of 4160 input prompts. Each image is a given rating between 1 and 5 (where 1 represents that *'image is irrelevant to the prompt'* and 5 represents that *'image is an accurate match for the prompt'*).

Furthermore, unlike prior works [7] which predominantly analyse the performance on relatively simple prompts with two subjects (*e.g.* object a and object b), we construct a systematically diverse pool of input prompts for better understanding text-to-image alignment across varying complexities in the text prompt. In particular, the prompts for the dataset are designed to encapsulate two axis of complexity: *number of subjects* and *realism*. The number of subjects refers to the number of main objects described in the input prompt and varies from 2 (*e.g.*, *a cat with a ball*) to 5 (*e.g.*, *a woman walking her dog on a leash by the beach during sunset*). Similarly, the *realism* of a prompt is defined as the degree to which different concepts naturally co-occur together and varies as *easy*, *medium*, *hard* and *very hard*. *easy* typically refers to prompts where concepts are naturally co-occurring together (*e.g.*, *a dog in a park*) while *very hard* refers to prompts where concept combination is very rare (*e.g.*, *a dog playing a piano*). Further details regarding the dataset are provided in supplementary material.

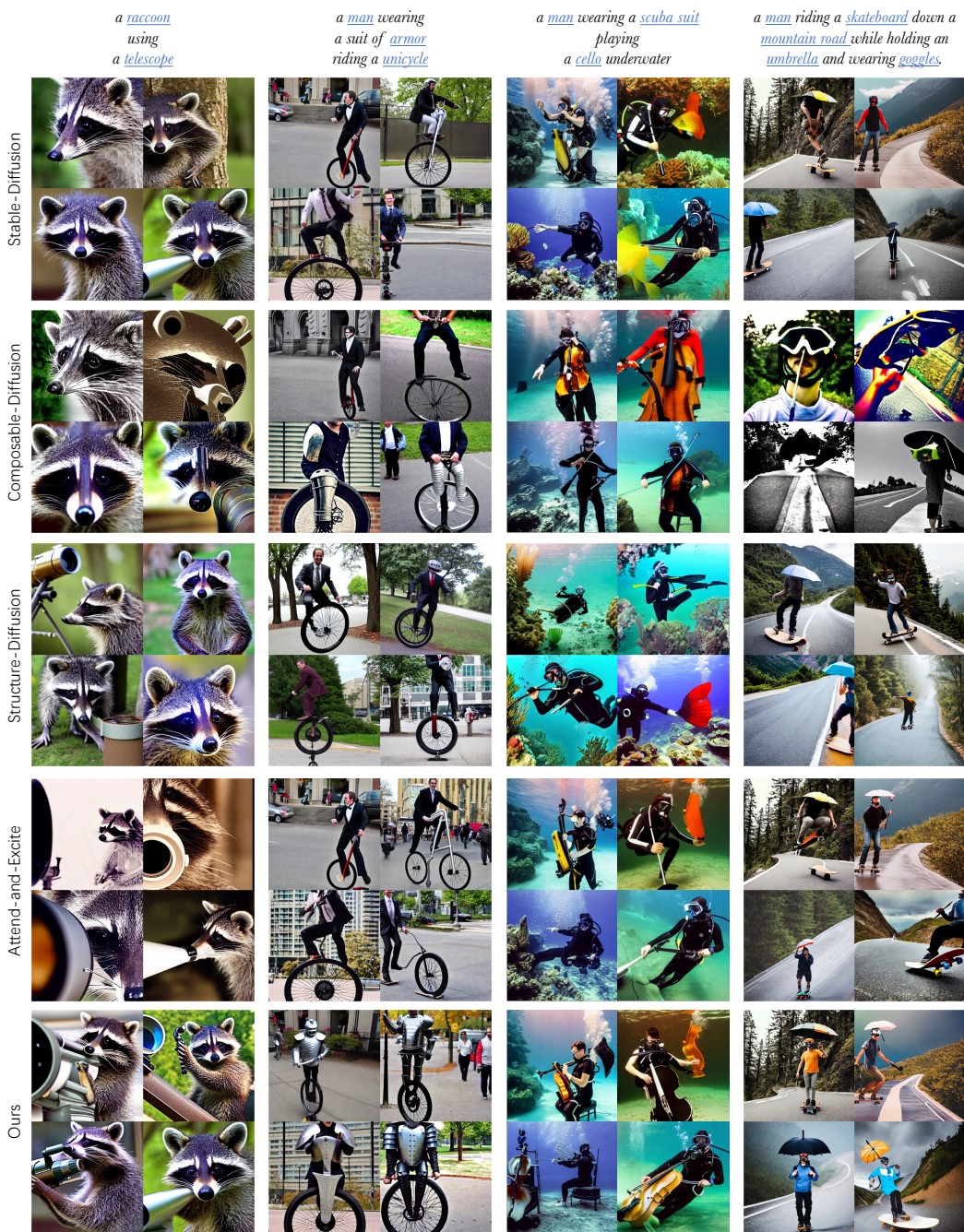

Figure 6: *Qualitative comparison w.r.t text-to-image alignment.* We compare the outputs of our iterative refinement approach with prior works [1, 6–8] on improving quality of generated images with changing number of subjects (underlined) from 2 to 5. Please zoom-in for best comparisons.

## 4.1 Evaluating Text-to-Image Alignment

**Baselines.** We compare the performance of the *Decompositional-Alignment Score* with prior works on evaluating text-to-image alignment in a reference-free manner. In particular, we show comparisons with CLIP [9], BLIP [10] and BLIP2 [11] scores where the text-to-image alignment score is computed using the cosine similarity between the corresponding image and text embeddings. We also include comparisons with BLIP-ITM and BLIP2-ITM which directly predict a binary image-text matching score (between 0 and 1) for input prompt and output image. Finally, we report results on the recently proposed text-to-text (T2T) similarity metric [7] which computes image-text similarity as the average cosine similarity between input prompt and captions generated (using BLIP) from the input image.

**Prompt**: A lion playing a guitar

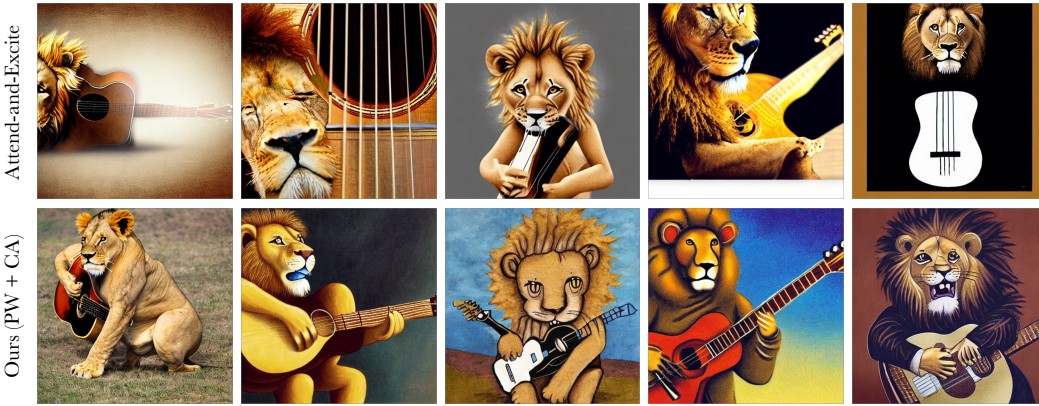

(a) *Object Relationship*: Eval-and-Refine helps better capture both presence and relationship between the objects.

**Prompt**: A person in a spacesuit riding a bicycle by the lake

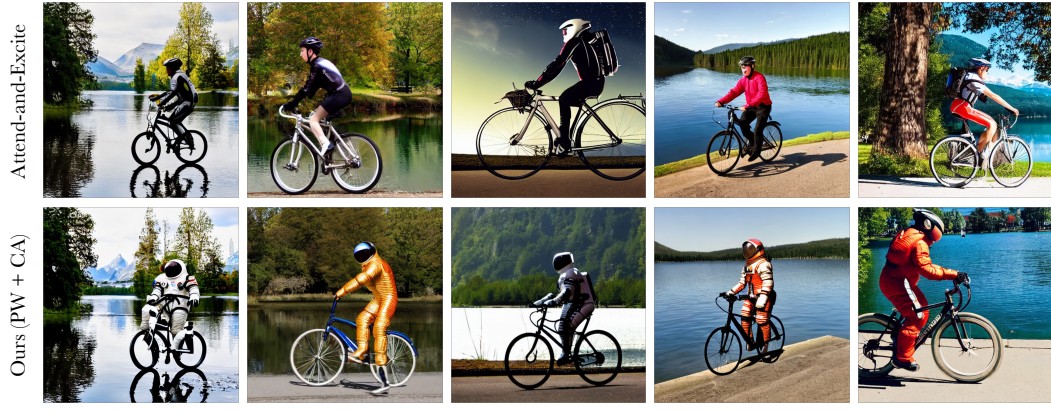

(b) *Overlapping entities*: Proposed approach can better handle cases with overlapping entities (spacesuit, person).

**Prompt**: a snowman wearing sunglasses holding an umbrella on a beach on a sunny day

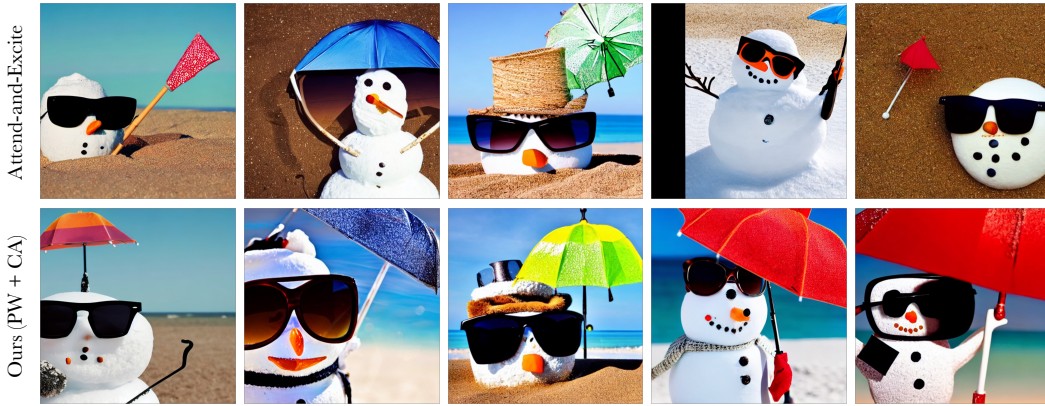

(c) *Prompt Complexity*: Eval-and-Refine shows better alignment as number of subjects in input prompt increase.

Figure 7: *Additional comparisons with Attend-and-Excite.* We analyse three main ways in which the proposed iterative-refinement improves over Attend-and-Excite [7] (refer Sec. 4.2 for details).

**Quantitative Results.** Fig. 5 shows the correlation between human annotations and predicted text-to-image alignment scores across different metrics on the *Decomposable-Captions* dataset. We observe that the *DA-Score* shows a significantly higher correlation with human evaluation ratings as opposed to prior works across varying number of subjects $N \in [2, 5]$ in the input prompt. We also note that

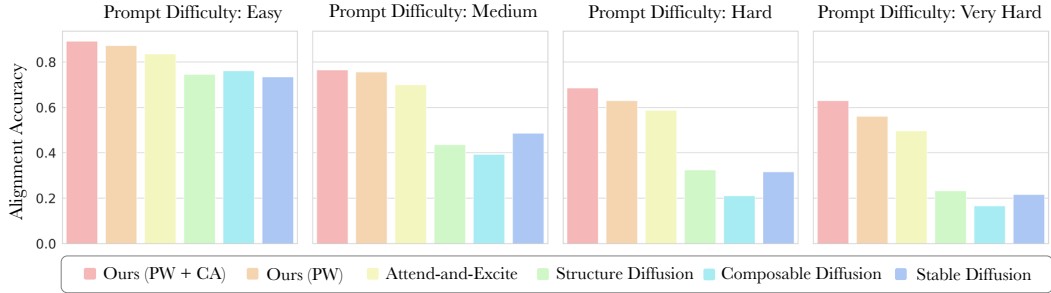

Figure 8: *Variation of alignment accuracy with prompt difficulty.* We observe that while the accuracy of all methods decreases with increasing difficulty in prompt *realism* (refer Sec. 4), the proposed iterative refinement approach consistently performs better than prior works.

while the recently proposed T2T similarity score [7] shows comparable correlation with ours for $N = 2$, its performance significantly drops as the number of subjects in the input prompt increases.

## 4.2 Improving Text-to-Image Alignment

In this section, we compare the performance of our iterative refinement approach with prior works on improving text-to-image alignment in a training-free manner. In particular, we show comparisons with 1) Stable Diffusion [1], 2) Composable Diffusion [6] 3) StructureDiffusion [8] and 4) Attend-and-Excite [7]. All images are generated using the same seed across all methods.

**Qualitative Results.** Results are shown in Fig. 6. As shown, we observe that Composable Diffusion [6] struggles to generate photorealistic combinations of objects especially as number of subjects in the prompt increase. StructureDiffusion [8] helps in addressing some missing objects *e.g.*, telescope in example-1, but the generated images tend to be semantically similar to those produced by the original Stable Diffusion model, and thus does not significantly improve text-to-image alignment.

Attend-and-Excite [7] shows much better performance in addressing missing objects (*e.g.*, telescope in example-1 and umbrella in example-4). However, as sumamrized in Fig. 7 we observe that it suffers from 3 main challenges: 1) *Object Relationship* (Fig. 7a): we observe that despite having desired objects, generated images may sometimes fail to convey relationship between them. For *e.g.*, in row-1 Fig. 7 while output images show both a *lion and guitar*, the lion does not seem to be playing the guitar. In contrast, Eval-and-Refine is able to describe both presence and relation between objects in a better manner. 2) *Overlapping Entities* (Fig. 7b): For images with overlapping entities (*e.g.*, *person and spacesuit*), we observe that Attend-and-Excite [7] typically spends most of gradient updates balancing between the overlapping entities, as both entities (*person and spacesuit*) occupy the same cross-attention region. This can lead to outputs where a) other important aspects (*e.g.*, *lake* in Col-3) or b) one of the two entities (*e.g.*, *spacesuit*) are ignored. 3) *Prompt Complexity* (Fig. 7c): Finally, we note that since Attend-and-Excite [7] is limited to applying the cross-attention update *w.r.t* the least dominant subject, as the complexity of input prompt $\mathcal{P}$ increases, it may miss some objects (*e.g.*, *umbrella, beach, sunny day*) during the generation process. In contrast, the iterative nature of our approach allows it to keep refining the output image $\mathcal{I}$ until a desirable threshold for the overall image-text alignment score $\Omega(\mathcal{I}, \mathcal{P})$ is reached.

**Quantitative Results.** In addition to qualitative experiments, we also evaluate the efficacy of our approach using human evaluations. In this regard, we report three metrics: 1) *normalized human score*: which refers to the average human rating (normalized between 0-1) for images generated on the Decomposable-Captions-4k dataset. 2) *accuracy*: indicating the percentage of generated images which are considered as an accurate match (rating: 5) for the input text prompt by a human subject. 3) *pairwise-preference*: where human subjects are shown pair of images generated using our method and prior work, and are supposed to classify each image-pair as a win, loss or tie (win meaning our method is preferred). For our approach we consider two variants 1) *Ours (PW)* which performs iterative refinement using only prompt-weighting, and 2) *Ours (PW + CA)* where iterative refinement is performed using both prompt weighting and introducing cross-attention updates (Sec. 3.2). Pairwise preference scores are reported while using *Ours (PW + CA)* while comparing with prior works.

| Method | Norm. Human Score (%) | Alignment Accuracy (%) | Pairwise Comparison % | | | Inference Time (s) |
|---|---|---|---|---|---|---|
| | | | Win ↑ | Tie | Lose ↓ | |
| Stable-Diffusion [1] | 72.98 | 43.66 | 41.7 | 50.1 | 8.2 | 3.54 s |
| Composable-Diffusion [6] | 70.28 | 37.72 | 57.1 | 38.5 | 4.4 | 10.89 s |
| Structure-Diffusion [8] | 74.93 | 45.23 | 37.5 | 54.6 | 7.9 | 11.51 s |
| Attend-and-Excite [7] | 85.94 | 65.50 | 23.6 | 62.3 | 14.1 | 8.59 s |
| Ours (PW) | 89.53 | 70.28 | N/A | N/A | N/A | 10.32 s |
| Ours (PW + CA) | **90.25** | **74.16** | N/A | N/A | N/A | 12.24 s |

Table 1: *Quantitative Results*. We report text-to-image alignment comparisons *w.r.t* normalized human rating score (Col:2), average alignment accuracy evaluated by human subjects (Col:3) and pairwise user-preference scores (ours vs prior work) (Col:4-6). Finally, we also report average inference time per image for different methods in Col:7. We observe that our approach shows better text-to-image alignment performance while on average using marginally higher inference time.

Results are shown in Fig. 8 and Tab. 1. We observe that while the text-to-image alignment accuracy for all methods decreases with an increased difficulty in input text prompts (Fig. 8), we find that the our approach with only prompt-weighting is able to consistently perform on-par or better than Attend-and-Excite [7]. Further introduction of cross-attention updates (Sec. 3.2), allows our approach to exhibit even better performance, which outperforms Attend-and-Excite [7] by 8.67 % in terms of overall alignment accuracy of the generated images. These improvements are also reflected in the pairwise comparisons where human subjects tend to prefer our approach over prior works [6–8].

**Inference time comparison.** Tab. 1 shows comparison for the average inference time (per image) for our approach with prior works [6–8]. We observe that despite the use of an iterative process for our approach, the overall inference time is comparable with prior works. This occurs because prior works themselves often include additional steps. For instance, Composable-Diffusion [6] requires the computation of separate denoising latents for each statement in the confunction/disjunction operation, thereby increasing the overall inference time almost linearly with number of subjects. Similarly, Attend-and-Excite [7] includes additional gradient descent steps for modifying cross-attention maps. Moreover, such an increase is accumulated even if the baseline Stable-Diffusion [1] model already generates accurate images. In contrast, the proposed iterative refinement approach is able to adaptively adjust the number of iterations required for the generation process by monitoring the proposed *DA-Score* for evaluating whether the generation outputs are already good enough.

## 5   Conclusion

In this paper, we explore a simple yet effective decompositional approach for both evaluation and improvement of text-to-image alignment with latent diffusion models. To this end, we first propose a *Decompositional-Alignment Score* which given a complex prompt breaks it down into a set of disjoint assertions. The alignment of each of these assertions with the generated image is then measured using a VQA model. The assertion-based alignment scores are finally combined to a give an overall text-to-image alignment score. Experimental results show that proposed metric shows significantly higher correlation with human subject ratings over traditional CLIP, BLIP based image-text matching scores. Finally, we propose a simple iterative refinement approach which uses the decompositional-alignment scores as feedback to gradually improve the quality of the generated images. Despite its simplicity, we find that the proposed approach is able to surpass previous state-of-the-art on text-to-image alignment accuracy while on average using only marginally higher inference times. We hope that our research can open new avenues for robust deployment of text-to-image models for practical applications.

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
