# Divide, Evaluate, and Refine: Evaluating and Improving Text-to-Image Alignment with Iterative VQA Feedback

## A  Additional Results

In this section, we include additional results for our approach which could not be included in the main paper due to space constraints. In particular, we report additional results for visualizing the iterative refinement process in Sec. A.1. We also provide additional results comparing our method performance with Attend-and-Excite [1] in Sec. A.2. Finally, in Sec. A.3, we compare our approach with recent contemporary work on using human feedback for improving text-to-image alignment.

### A.1  Visualizing the Iterative Refinement Process

a man wearing a scuba suit playing a cello underwater with fish swimming around

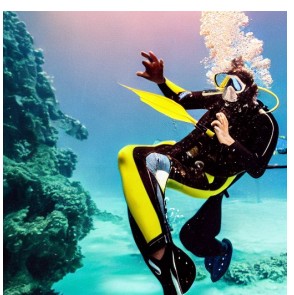 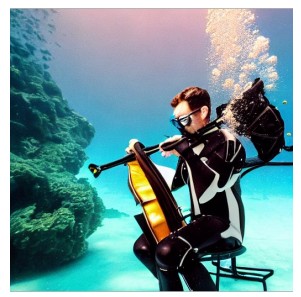 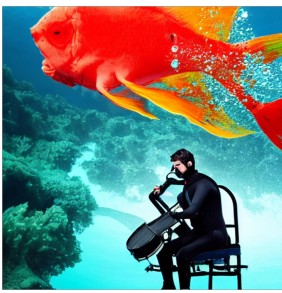

a penguin wearing a bowtie standing on a surfboard in a swimming pool

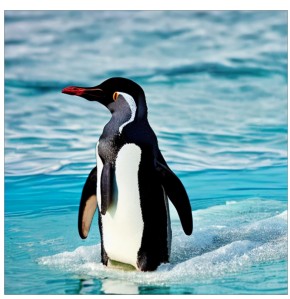 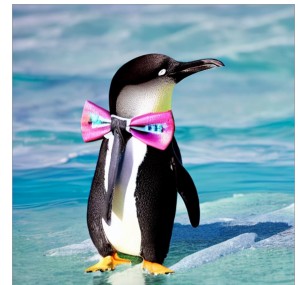 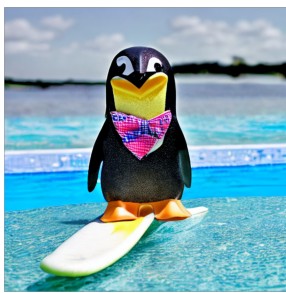

a fish jumping out of the water to catch a butterfly near a waterfall

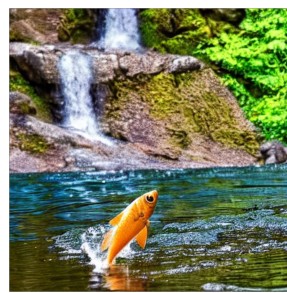 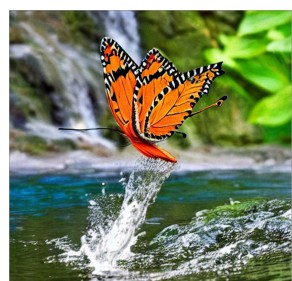 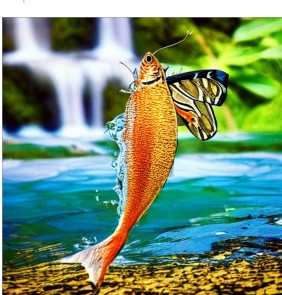

Figure 1: *Visualizing iterative refinement process for improving text-to-image alignment.*

a cat wearing a life jacket on a canoe going down a river

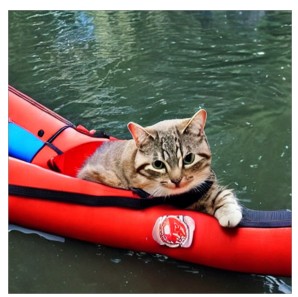 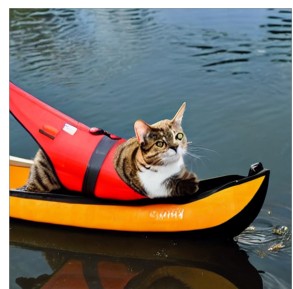 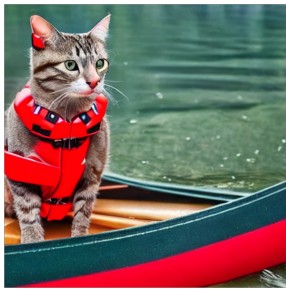

a monkey wearing a suit giving a presentation in a conference room

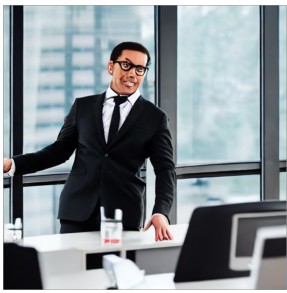 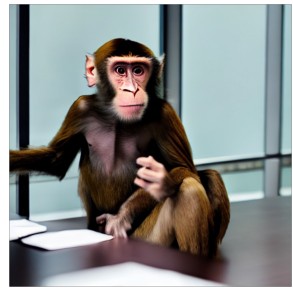 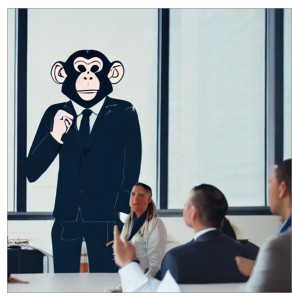

a dolphin wearing a graduation cap in a pool holding a diploma

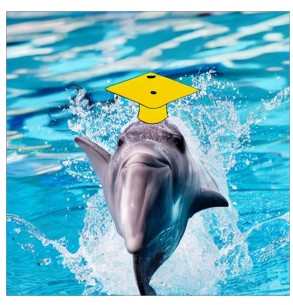 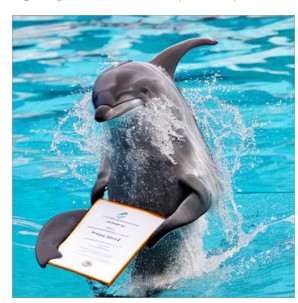 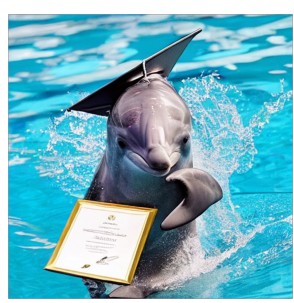

a bird wearing a tiny hat on a branch next to a small cafe

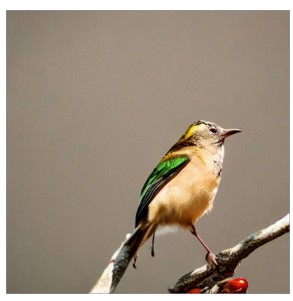 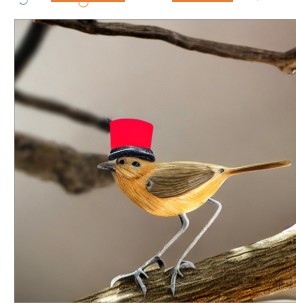 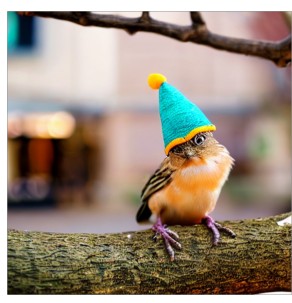

a pod of dolphins jumping out of the water in an ocean with a ship in the background

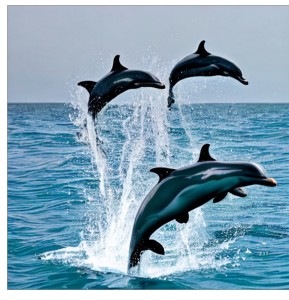 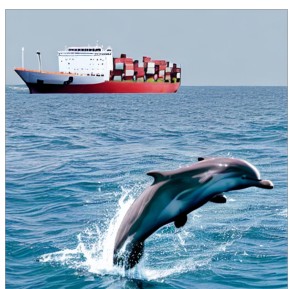 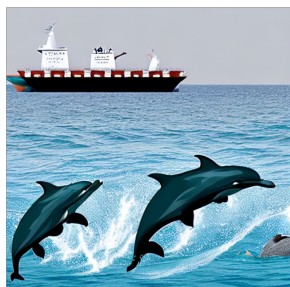

Figure 2: *Visualizing iterative refinement process for improving text-to-image alignment.*

## A.2 Additional Comparisons with Attend-and-Excite

Prompt: A lion playing a guitar

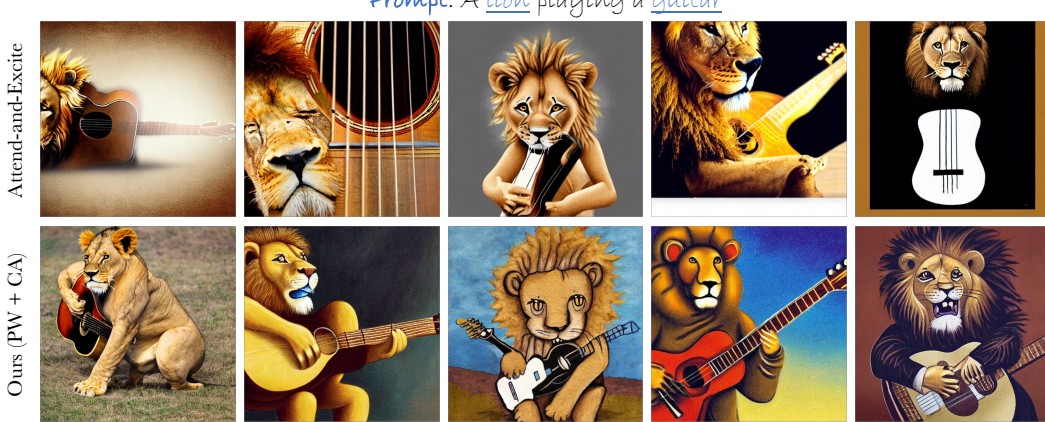

Figure 3: *Additional Comparisons with Attend-and-Exite: Missing Relationship.* Due to the pure object focused nature of Attend-and-Excite [1], it may result in images where all objects are present but the relationship between them is not accurately described. In contrast, we observe that the iterative refinement approach is able to better describe both presence and relationship between the objects.

Prompt: A person in a spacesuit riding a bicycle by the lake

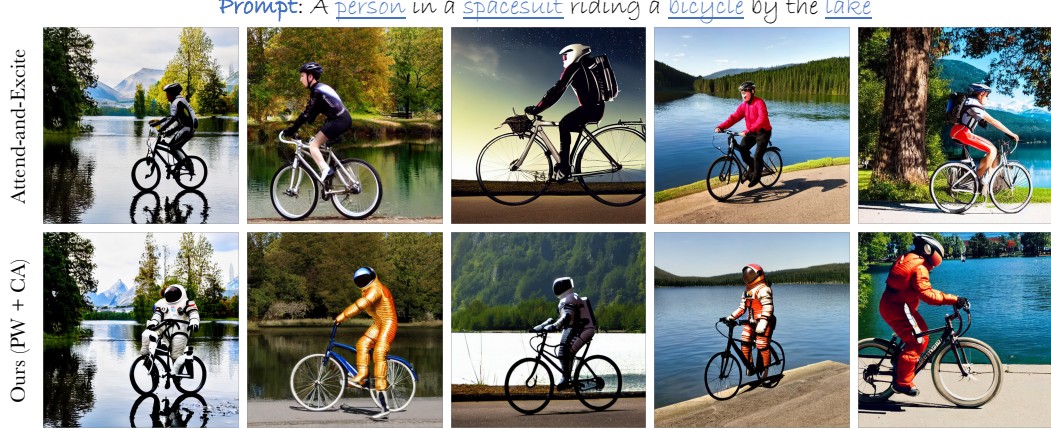

Figure 4: *Additional Comparisons with Attend-and-Exite: Overlapping Entities.* For images with overlapping entities (*e.g.*, *person and spacesuit*), we observe that Attend-and-Excite [1] typically spends most of gradient updates balancing between the overlapping entities, as both entities (*person and spacesuit*) occupy the same cross-attention region. This can lead to outputs where a) other important aspects (*e.g.*, *lake* in Col-3) or b) one of the two entities (*e.g.*, *spacesuit*) are ignored.

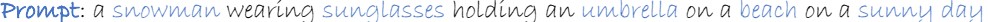

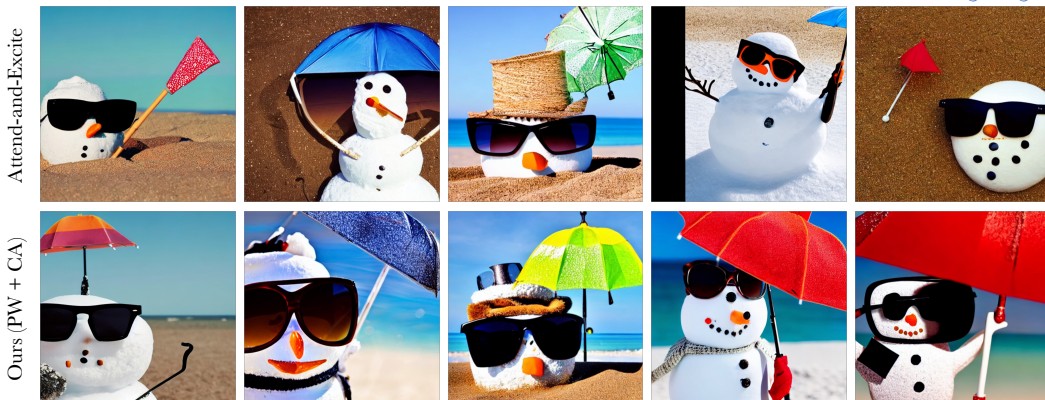

Figure 5: *Additional Comparisons with Attend-and-Exite: Missing Objects.* Since Attend-and-Excite [1] is limited to applying the cross-attention update *w.r.t* the least dominant subject, as the complexity of input prompt increases, it may miss some objects (*e.g.*, umbrella, beach, sunny day) during the generation process. In contrast, the iterative nature of our approach allows it to keep refining the output image until a desirable threshold for the overall image-text alignment score is reached.

## A.3 Comparisons with Human-Feedback based Methods

Besides training-free methods, recent contemporary work [2, 4] has also explored the possibility of improving image-text alignment using human feedback to finetune existing latent diffusion models. For instance, Wu *et al*. [2] recently release new versions of CLIP [3] and Stable Diffusion [5] models which have been finetuned to better align with user preferences using human-feedback data. In this section, we compare the performance of our simple training-free approach with models released by [2] in terms of both evaluation and improvement of fine-grain text-image matching.

**Results.** Fig. 6 compares the correlation of 1) Original CLIP [3] model, 2) HPS scores (CLIP model finetuned to better align with human preference scores) by [2], and 3) DA-Scores predicted by our method on the Decomposable-Captions-4k dataset. We observe that while the HPS scores with the finetuned CLIP model, show significantly higher correlation with human annotations as opposed to original CLIP model, it still performs worse than the proposed DA-scores.

Similarly, Fig. 7 compares the image outputs for 1) Original Stable Diffusion [5] model, 2) HPS Adapted Stable Diffusion model from [2] and 3) DA-score based iterative-refinement approach. We observe that while the HPS Adapted model improves the aesthetics of the generated model (*e.g.*, improved lighting in Col-1,5 in example-1 Fig. 7), it does not improve the semantic alignment with content of the input prompt. In contrast, while our approach does not improve the aesthetics of the generated image, the output images show significantly higher alignment with the input prompt.

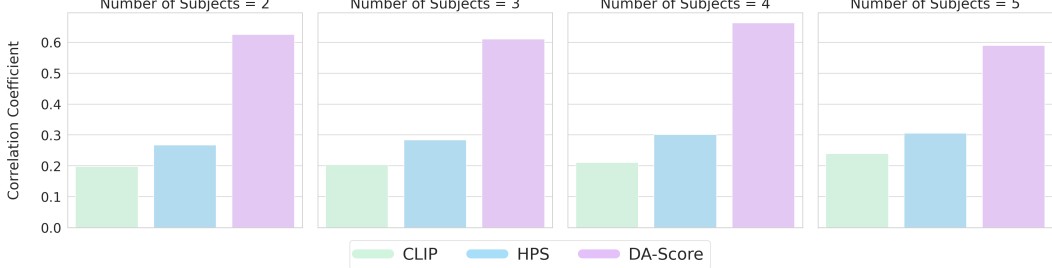

Figure 6: *Method comparisons w.r.t correlation with human ratings.* We compare the correlation of HPS score (CLIP finetuned with human feedback) from Wu *et al*. [2]. We observe that while the HPS score shows improved performance over CLIP [3], the proposed DA-score shows better correlation with human ratings across varying number of subjects in input prompt.

Prompt: A man wearing scuba suit playing a cello underwater

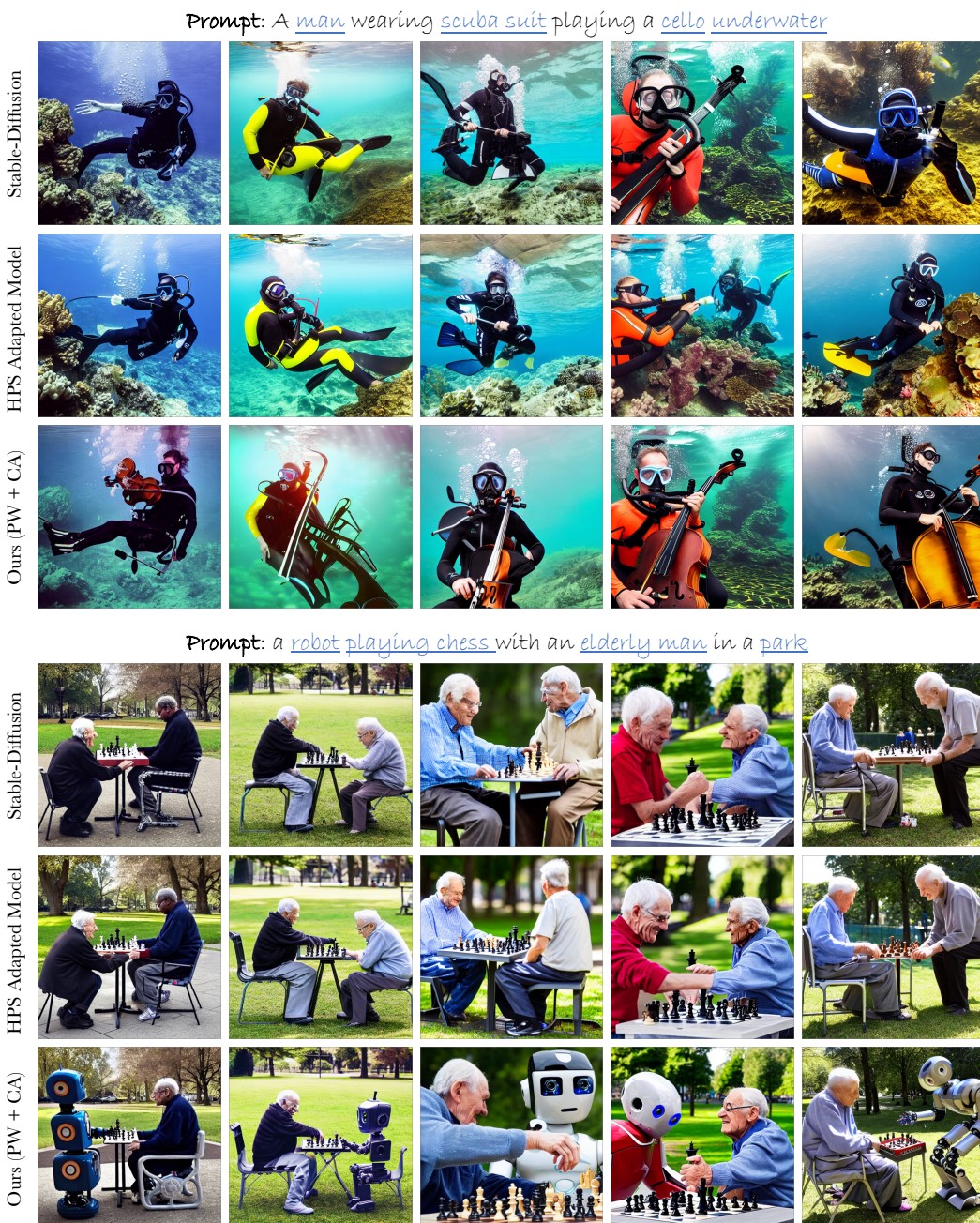

Prompt: a robot playing chess with an elderly man in a park

Figure 7: *Comparing image alignment performance with human-feedback based models.* We find that while human-feedback based finetuned diffusion model from [2] improves the aesthetics (*e.g.* lighting) of the output, it does not visually improve alignment with the input text for complex prompts.

**Reason.** As shown above while human preference finetuned CLIP and Stable Diffusion models from [2] show better performance in terms of aesthetics, they do not significantly improve visual alignment between generated images and input text as the complexity of the prompts increases. *We believe that a major reason behind the same is the heavy data-driven nature of human-feedback based methods. That is, the generalization performance of the final finetuned model often relies heavily on the diversity and nature of the human-feedback dataset.* In current works [2, 4], human-feedback is typically collected by showing users $4 - 10$ images (for the same input prompt) generated by a pretrained Stable-Diffusion [5] model, and then asking the users to select the best match. However as shown in

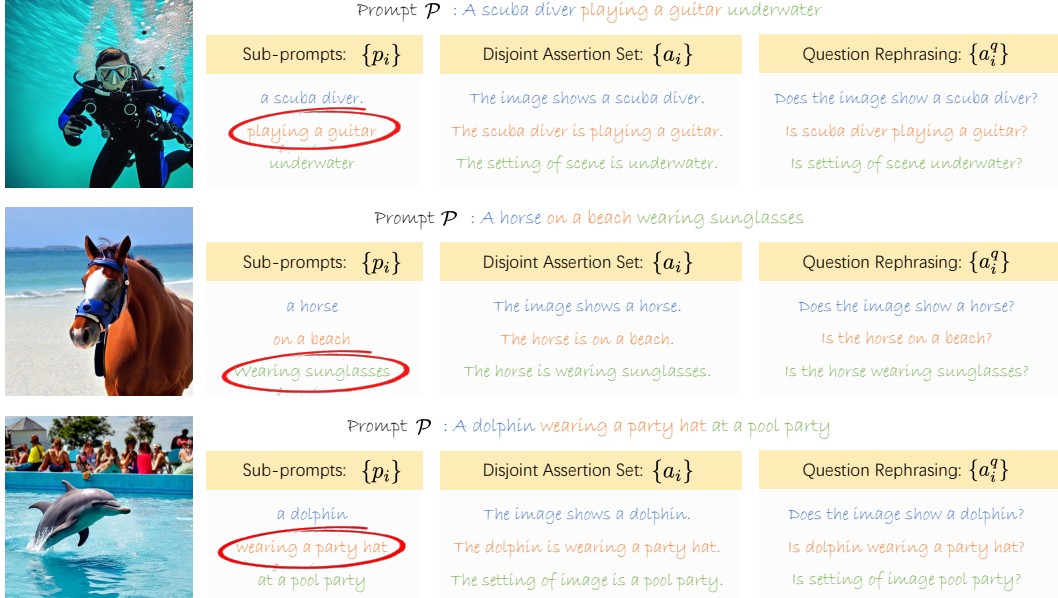

Figure 8: *Visualizing the outputs of prompt decomposition.* By dividing a complex prompt $\mathcal{P}$ into a set of disjoint assertions $a_i$, we are able to identify the sub-prompts $p_i$ (circled) which are not being expressed in the image output using VQA, and thereby address them using iterative refinement.

the main paper, as the complexity of the input prompt increases, the original Stable-Diffusion [5] model shows a very low text-to-image aligment accuracy. As a result, the collected human-feedback data is often biased towards predicting more aesthically pleasing outputs, as opposed to outputs which improve fine-grain alignment with the content of the input prompt.

In contrast, we propose a simple training-free approach which is able to generalize well to both simple and hard prompts, for both evaluation and improvement of text-to-image alignment. We also note that training-free methods such as ours or prior works [1, 6, 7], can in-turn help improve the performance of human-feedback based methods by providing a better quality dataset for determining both aesthetics and content alignment of the generated images.

## B  Implementation Details

In this section, we provide further details for the implementation of our approach as well as other baselines [1, 5–7] used while reporting results in the main paper. The full detailed implementation for both evaluation and improvement of text-to-image alignment is provided in Alg. 1, 2.

**Model Details.** Similar to [1, 7], we use the official Stable Diffusion v1.4 model as the underlying pretrained text-to-image generation model while reporting results with all methods [1, 6, 7] (including the iterative refinement approach proposed in the main paper). All results are reported at $512 \times 512$ resolution while using 50 inference steps during the reverse diffusion process. Unless otherwise specified, a fixed classifier-free guidance scale [8] $\alpha_{cfg} = 7.5$ is used for all experiments. By default, we use the pretrained BLIP-VQA [9] model (BLIP model finetuned for visual question answering) for predicting the assertion-alignment scores while reporting all results.

**Prompt Decomposition.** Similar to our approach, prior works on improving image-text alignment often rely on human-user inputs for expressing contents of the input prompt into its simpler constituents. For instance, Feng *et al.* [6] require the user to describe the prompt as a conjunction / disjunction of simpler statements. Chefer *et al.* [1] require the user to provide a set of entities / subjects in the prompt, over which the cross-attention optimization should be performed. Similarly, in order to evaluate the *Decompositional Alignement Scores*, our approach relies on decomposing the input caption into a set of disjoint assertions (with their rephrasing as a question).

Instead of relying of human inputs as in prior works, we leverage the in-context learning capability of large-language models (LLMs) [10, 11] for obtaining such decompositions in an autonomous

manner. This allows us to perform large-scale quantitative evaluations across different methods in a robust manner. In particular, for all methods [1, 6] and ours, we first collect a set of 4-5 human generated examples describing the desired outputs (*e.g.*, main subjects for [1]) for the input prompts. This example-set is then used as an in-context dataset which is then fed to the *'gpt-4'* [10] model to generate the desired prompt decompositions on prompts across the *Decomposable-Captions-4k* dataset. Fig. 8 provides an overview of the different prompt decomposition outputs for our approach.

Note that while it is possible to explore relatively simpler methods *e.g.*, extracting noun phrases for subject extraction in [1], it can lead to errors as the complexity of the input prompt increases. For instance, for the input *prompt: "a penguin wearing a bowtie with a bright sun in the background"*, the extracted noun phrases will also include the word *"background"*. The use of a LLM-based in-context framework allows us to avoid such errors. In future, the proposed approach can be extended to allow for much lightweight decomposition by using a low-rank finetuning [12] for adaptation of recently released instruction-following models [13]. However, since the same is not the main focus of our work, we leave it as a directive for future research.

**Hyperparameters and Overall Algorithm.** The proposed iterative refinement approach uses a maximum of $K = 5$ iterations for the refinement process. The iterative process is terminated early if a threshold of 0.8 for the overall image alignment score $\Omega(\mathcal{I}_k, \mathcal{P})$ is obtained. The iterative refinement weights are initialized as $w_i = 1 \forall i$ for prompt weighting (PW), and, $\gamma_i = 0 \forall i$ for cross-attention (CA) updates. An increment $\Delta$ of 0.1 and 1.0 is used for updating the assertion weights for prompt-weighting (PW) and cross-attention (CA) update methods respectively. Furthermore, to reduce the inference time for each iteration as compared to [1], cross-attention updates are only applied for first 20 steps of the reverse diffusion process. Furthermore, the use of iterative cross attention updates is also discarded. The image generation output $\mathcal{I}^\star$ at the end of the refinement process is computed as,

$$\mathcal{I}^\star = \operatorname{argmax}_{\mathcal{I}_k} \Omega(\mathcal{I}_k, \mathcal{P}). \tag{1}$$

Please refer Alg. 1, 2 for the full detailed implementation (with hyperparameters) for our approach.

---

**Algorithm 1** DA-Score: Evaluating Text-to-Image Alignment

---

**Input**: Text prompt $\mathcal{P}$, generated image $\mathcal{I}$.
**Output**: Text-to-Image Alignment Score between $\mathcal{P}, \mathcal{I}$
**Require**: Large-language model $\mathcal{M}$, VQA-model $\mathcal{V}$, exempler dataset $\mathcal{D}$, task description $\mathcal{T}$, softmax-temperature $\tau = 0.9$

1: ▷ PROMPT DECOMPOSITION
2: $\mathbf{x} = \{x_0, x_1, \ldots x_n\} = \mathcal{M}(\mathbf{x} \mid \mathcal{P}, \mathcal{D}_{exempler}, \mathcal{T}), \quad where \ \ x_i = \{a_i, p_i, a_i^q\};$
3:
4: ▷ COMPUTE ASSERTION ALIGNMENT SCORES USING VQA
5: $u_i(\mathcal{I}, a_i) = \frac{\exp{(\alpha_i/\tau)}}{\exp{(\alpha_i/\tau)} + \exp{(\beta_i/\tau)}}, where \ \ \alpha_i = \mathcal{V}(\text{'yes'} \mid \mathcal{I}, a_i^q), \quad \beta_i = \mathcal{V}(\text{'no'} \mid \mathcal{I}, a_i^q)$
6:
7: ▷ OVERALL IMAGE-TEXT ALIGNMENT SCORE
8: $\Omega(\mathcal{I}, \mathcal{P}) = \sum_i \lambda_i(\mathcal{P}, a_i) \, u_i(\mathcal{I}, a_i) / \sum_i \lambda_i(\mathcal{P}, a_i),$
9:
10: **return** $\Omega(\mathcal{I}, \mathcal{P}).$

---

## C Decomposable Captions 4K Dataset

**Overview.** Since there are no openly available datasets addressing semantic challenges in text-based image generation with human annotations, we introduce a new benchmark dataset Decomposable-Captions-4k for method comparison. The dataset consists an overall of 24960 human annotations on images generated using all methods [1, 5, 6] (including ours) across a diverse set of 4160 input prompts. Each image is a given rating between 1 and 5 (where 1 represents that *'image is semantically irrelevant to the prompt'* and 5 represents that *'image is an accurate match for the prompt'*). Fig. 9 provides an overview of some user annotations for image-prompt pairs from the curated dataset.

**Collecting Diverse Prompts of Varying Complexity.** Furthermore, unlike prior works [1] which predominantly analyse the performance on relatively simple prompts with two subjects (*e.g.*, object a and object b), we construct a systematically diverse pool of input prompts for better understanding

---

**Algorithm 2** Iterative Refinement: Improving Text-to-Image Alignment

---

**Input**: Text prompt $\mathcal{P}$, subprompts $p_i$, disjoint assertions in question format $a_i^q$.
**Output**: Image generation output $\mathcal{I}^\star$ conditioned on $\mathcal{P}$
**Require**: Pretrained diffusion model $\mathcal{D}$, VQA-model $\mathcal{V}$, prompt-weighting function $\mathcal{W}$
**Hyperparameters**: max number of iterations $K = 5$, alignment threshold $\Omega_{max} = 0.8$, weight increments $\Delta_w = 0.1$, $\Delta_\gamma = 1.0$, step-size $\alpha = 10$, number of cross-attention update steps $t_0 = 20$.

1:   ▷ INITIALIZE ASSERTION WEIGHTS
2:   Initialize $w_i^0 = 1 \; \forall i$;                                               ▷ for prompt-weighting
3:   Initialize $\gamma_i^0 = 0 \; \forall i$;                                           ▷ for cross-attention updates
4:
5:   ▷ ITERATIVE REFINEMENT
6:   **for** $0 \leq k < K$ **do**
7:
8:       ▷ PROMPT WEIGHTING
9:       $y_{prompt} = \mathcal{W}(\mathcal{P}, \{\text{CLIP}(p_i), w_i^k\}_{i=1}^n)$;
10:
11:       ▷ PARAMETERIZED REVERSE DIFFUSION
12:       Sample $z_T \sim \mathcal{N}(0, \boldsymbol{I})$;
13:       **for** $0 < t \leq T$ **do**
14:           $\_, \mathcal{A}_i^t = \mathcal{D}(z_t, y_{prompt}, t)$;                   ▷ compute cross-attention maps
15:           **if** $t \geq T - t_0$ **then**
16:               ▷ WEIGHTED CROSS-ATTENTION UPDATES
17:               $\mathcal{L}(z_t, \{\gamma_i^k\}_{i=1}^n) = \sum_i \gamma_i^k (1 - \max G(\mathcal{A}_i^t))$;
18:               $z_t = z_t - \alpha \nabla_{z_t} \mathcal{L}(z_t, \{\gamma_i^k\}_{i=1}^n)$;
19:           **end if**
20:           $z_{t-1} = \text{REVERSEDIFF}(z_t, t \rightarrow t - 1 \mid y_{prompt})$;
21:       **end for**
22:
23:       ▷ GET DECOMPOSITIONAL-ALIGNMENT SCORES
24:       $\mathcal{I}_k = x_0 = \text{DECODER}(z_0)$;
25:       $\Omega(\mathcal{I}_k, \mathcal{P}), \{u_i(\mathcal{I}_k, \mathcal{P})\}_i^n = \text{DA-SCORE}(\mathcal{I}_k, \{a_i^q\}_i)$;
26:
27:       ▷ FINISH IF OUTPUT IS GOOD ENOUGH
28:       **if** $\Omega(\mathcal{I}_k, \mathcal{P}) \geq \Omega_{max}$ **then**
29:           **return** $\mathcal{I}^\star = \mathcal{I}_k$.
30:       **end if**
31:
32:       ▷ UPDATE ASSERTION WEIGHTS
33:       $w_i^{k+1} = \begin{cases} w_i^k + \Delta_w, & \text{if} \quad i = \text{argmin}_l \, u_l(\mathcal{I}_k, \mathcal{P}) \\ w_i^k & \text{otherwise} \end{cases}$          ▷ for prompt-weighting
34:       $\gamma_i^{k+1} = \begin{cases} \gamma_i^k + \Delta_\gamma, & \text{if} \quad i = \text{argmin}_l \, u_l(\mathcal{I}_k, \mathcal{P}) \\ \gamma_i^k & \text{otherwise} \end{cases}$       ▷ for cross-attention updates
35: **end for**
36:
37: ▷ RETURN BEST OUTPUT
38: **return** $\mathcal{I}^\star = \text{argmax}_{\mathcal{I}_k} \Omega(\mathcal{I}_k, \mathcal{P})$.

---

text-to-image alignment across varying complexities in the text prompt. In particular, the prompts for the dataset are designed to encapsulate two axis of complexity: *number of subjects* and *realism*. The number of subjects refers to the number of main objects described in the input prompt and varies from 2 (*e.g.*, *a cat with a ball*) to 5 (*e.g.*, *a woman walking her dog on a leash by the beach during sunset*). Similarly, the *realism* of a prompt is defined as the degree to which different concepts naturally co-occur together and varies as *easy*, *medium*, *hard* and *very hard*. *easy* typically refers to prompts where concepts are naturally co-occurring together (*e.g.*, *a dog in a park*) while *very hard* refers to prompts where concept combination is very rare (*e.g.*, *a dog playing a piano*).

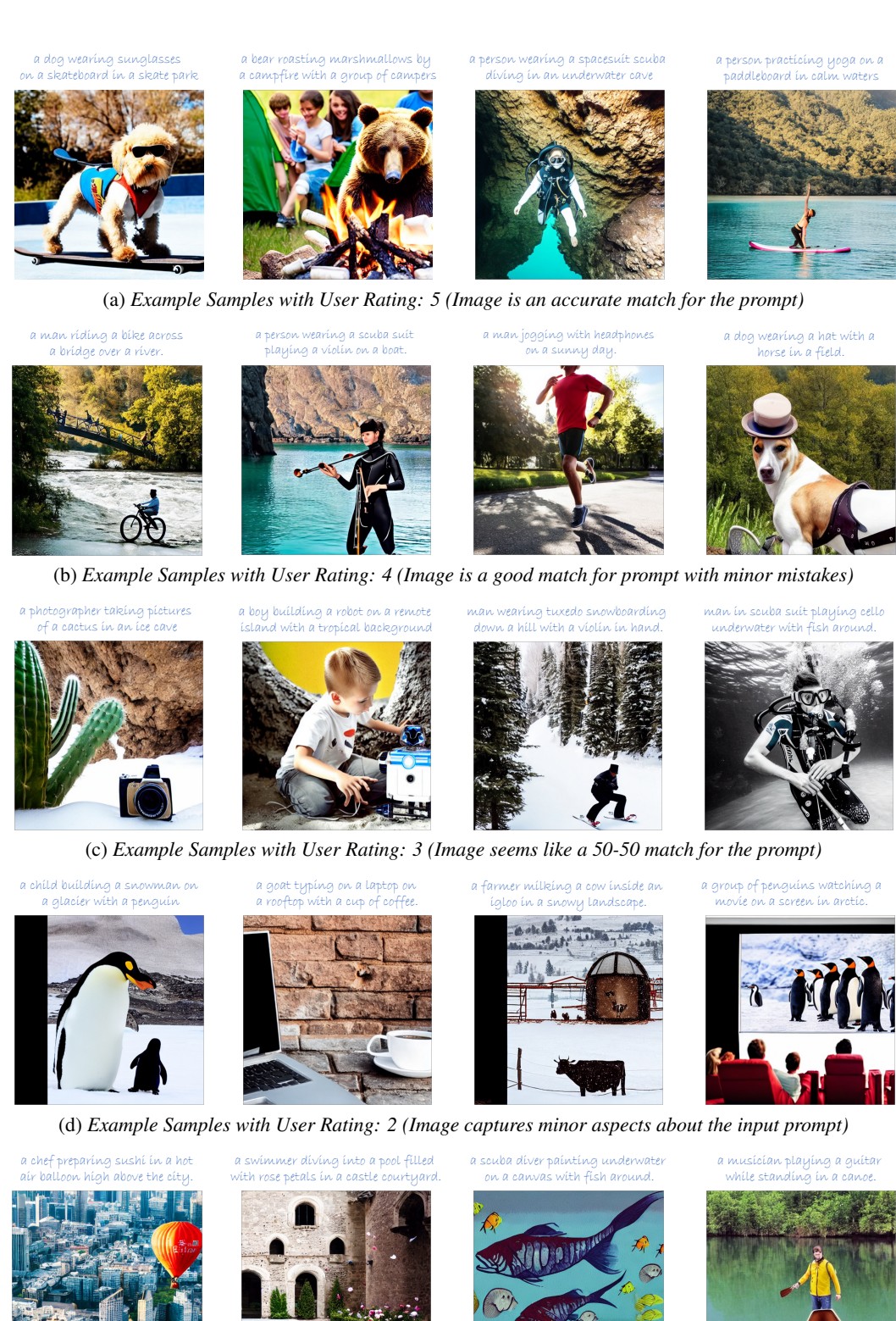

(a) *Example Samples with User Rating: 5 (Image is an accurate match for the prompt)*

(b) *Example Samples with User Rating: 4 (Image is a good match for prompt with minor mistakes)*

(c) *Example Samples with User Rating: 3 (Image seems like a 50-50 match for the prompt)*

(d) *Example Samples with User Rating: 2 (Image captures minor aspects about the input prompt)*

(e) *Example Samples with User Rating: 1 (Image seems semantically irrelevant to the prompt)*

Figure 9: *Visualizing samples with human annotations from the Decomposable-Captions-4k dataset.*

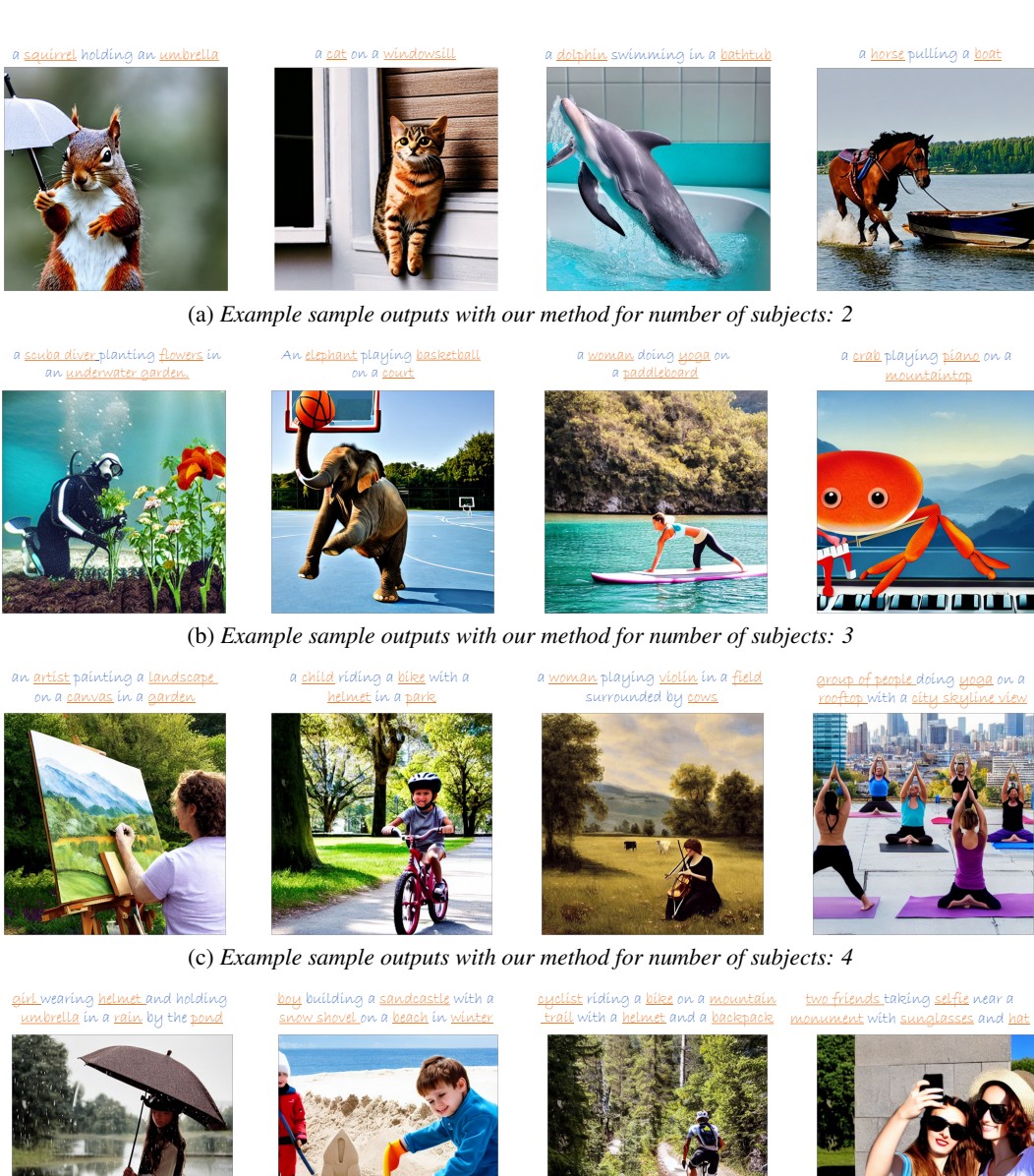

(a) *Example sample outputs with our method for number of subjects: 2*

(b) *Example sample outputs with our method for number of subjects: 3*

(c) *Example sample outputs with our method for number of subjects: 4*

(d) *Example sample outputs with our method for number of subjects: 5*

Figure 10: *Visualizing variation in number of subjects in prompts from the Decomposable-Captions-4k dataset. All images are generated using the proposed iterative refinement approach.*

**Prompt Generation.** A key part of the *Decomposable-Captions-4k* dataset is to collect a set of diverse input prompts of varying complexity which would allow for a much more comprehensive evaluation across different methods. Moreover unlike prompts found in typical large-scale image-text datasets [14, 15], the generated prompts should be imaginative and be able to describe novel and often non-realistic combinations of different concepts (*e.g. a lion playing a piano*).

To this end, we leverage the diverse language modelling capabilities of large-scale large language models (LLM) [10, 11] in order to generate novel prompts of varying complexity and realism. In particular, given a desired number of subjects $N$ (between 2 and 5), we first use the *GPT-4* model [10] API with 8K context length to come up with an initial random subject *e.g.*, a dog. The model is then instructed to conditionally generate a second subject (*e.g.*, sunglasses) which is then combined with

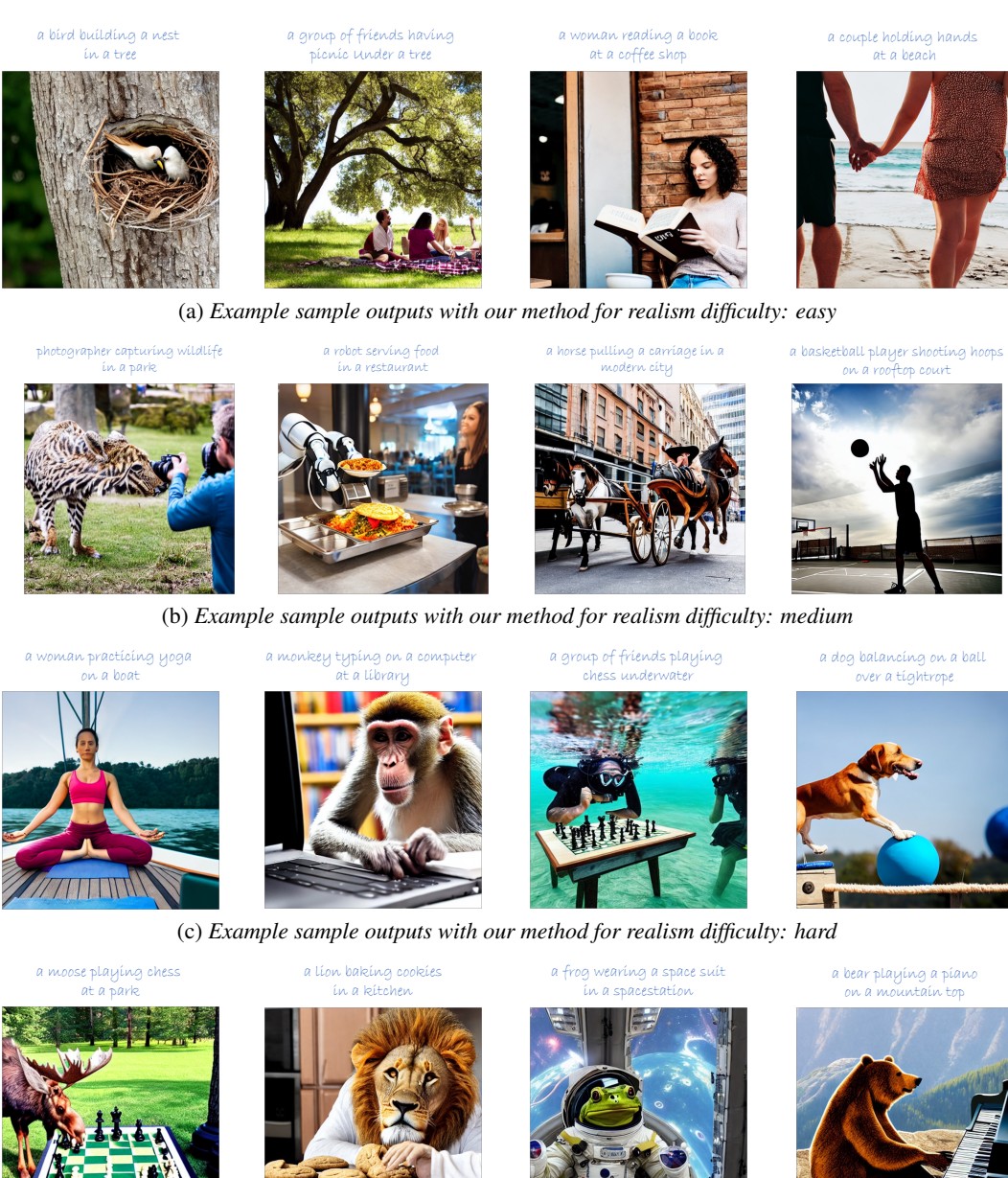

(a) *Example sample outputs with our method for realism difficulty: easy*

(b) *Example sample outputs with our method for realism difficulty: medium*

(c) *Example sample outputs with our method for realism difficulty: hard*

(d) *Example sample outputs with our method for realism difficulty: very hard*

Figure 11: *Visualizing variation in realism difficulty from the Decomposable-Captions-4k dataset. All prompts have been sampled using number of subjects = 3 subset of the overall dataset.*

first subject to generate the sub-prompt *"a dog wearing sunglasses"*. This process is continued until a complete prompt with a desired number of subjects is obtained (*e.g., a dog wearing sunglasses on a skateboard in a park*). At the end of generation process, prompts which are grammatically inaccurate are filtered and removed. An overview of example prompts with variable number of subjects (along with corresponding image generation outputs with our approach) is provided in Fig. 10.

Furthermore, in order to generate prompts of varying level of *realism difficulty*, we generate prompts in batches of 4. In particular, during the sequential generation process (described above) the model is prompted to generate prompts of increasing level of *realism difficulty* by asking it generate additional subjects whose combination in a sentence is increasingly more rare. For instance, for *realism difficulty: easy*, the model is tasked to generate additional subjects which typically co-occur together

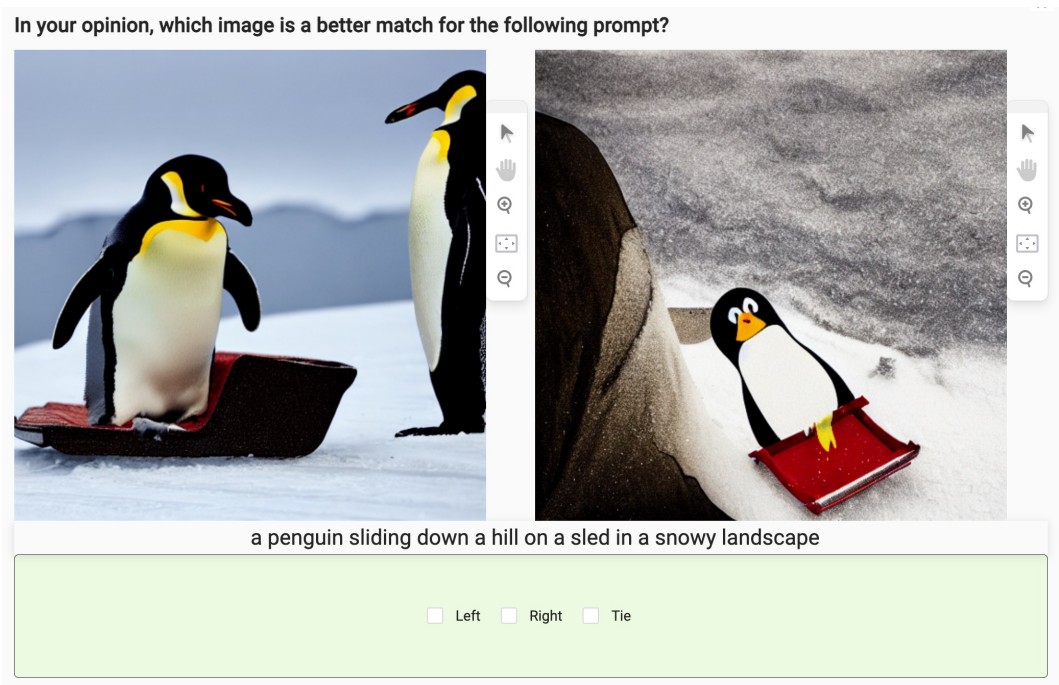

Figure 12: *Setup for pairwise user study comparing our method with prior works.*

in natural captions, which then leads to natural realistic prompts (*e.g.*, *a group of friends having picnic under a tree*). As the *realism difficulty* is increased the model generates input prompts where the co-occurence of different subjects is more and more rare, thus allowing it to generate more imaginative and challenging prompts (*e.g. a lion baking cookies in a kitchen*). Fig. 11 provides an overview of some example prompts with varying levels of realism difficulty.

**Pairwise Human User Study.** In addition to obtaining human annotations rating (between 1 to 5) for each image-prompt pair (refer Fig. 9), we also perform a pairwise user study comparing our method with prior works. In particular, given an input text prompt $\mathcal{P}$, the participants were shown a pair of image generation outputs comparing our method with prior works. For each pair, the human subject is then asked to select the output image which better aligns with the input prompt description. The human subjects are given three options *left*, *right* and *tie*, where *tie* indicates that both images are equally good or bad. All comparison images are generated using the same seed at $512 \times 512$ resolution with 50 inference steps for the reverse diffusion process. Fig. 12 provides a screenshot of the user interface for collecting the above human annotation data with pairwise comparisons.

## D    Discussion and Limitations

While the proposed iterative refinement approach shows better performance than previous works [1, 6, 7], it still has some limitations. *First*, the proposed decompositional approach relies on a pretrained BLIP-VQA model [9], for determining the alignment of the generated image with each of the disjoint assertions. Thus, weaknesses of the pretrained BLIP-VQA model become our weaknesses. Recall that the VQA scores help identify the areas in which the current image generation output is lacking, which can then be addressed in the next refinement step. However, if the VQA output is not correct, then the model might focus on assertions which are already well expressed, which can reduce the efficiency of the proposed iterative refinement strategy. In future, the use of more accurate VQA models *e.g.*, BLIP2-VQA can help alleviate this problem. Furthermore, as noted through extensive quantitative experiments across a diverse range of input prompts (refer main paper and App. C), we find that the use of the BLIP-VQA model still yields quite competitive results.

*Second*, without additional information from the user, the iterative refinement approach considers all assertions to be equally important in determining the overall content of the input prompt. However, as

the complexity of the input prompt increases, we may wish to give more weight to certain parts of the prompt over the others. Furthermore, it is possible that the prompt may contain assertions which are not visually verifiable from the output image. For instance, for the prompt *"a penguin in a shopping mall on a weekend"*, the assertion about whether it is a weekend or not is not visually verifiable. Similarly certain actions *e.g.* searching, singing are also not verifiable from a single image. In future, we would like to explore a more autonomous mechanism for including additional information like 1) user ranking for different assertions (*i.e.*, what the user considers as important in a prompt) as well as 2) visual verifiability of a given assertion while computing the decompositional alignment scores.

*Finally,* as noted in Fig. 7 of the main paper, we note that while the proposed iterative refinement approach leads to consistent improvements in alignment accuracy over prior works, the accuracy of the alignment process decreases as the complexity of input prompt is increased. In particular, for prompts with *very hard* realism difficulty, the overall alignment accuracy is only 62.9% (Attend-and-Excite has 49.5%). This leaves much room for improvement of text-to-image generation models. As discussed in App. A.3 one potential solution in this direction would be to combine recent works on human-feedback based diffusion model finetuning with the proposed training-free approach for data collection. In particular, by generating training data (on which human feedback is obtained) using the proposed iterative refinement strategy instead of previously used pretrained Stable Diffusion [5] models, we can increase the quality of the finetuning process. Using the proposed decompositional alignment scores as pseudo-labels for learning the human-feedback based reward model [4] is another interesting direction for future work. However, the same is out of scope of this paper, and we leave it as a direction for future research.