# OpenReview forum: "Divide, Evaluate, and Refine: Evaluating and Improving Text-to-Image Alignment with Iterative VQA Feedback"
_NeurIPS.cc/2023/Conference — NeurIPS 2023 poster_

### Official Review · Reviewer_5iW3 · 2023-07-04

**Soundness:** 4 excellent
**Presentation:** 4 excellent
**Contribution:** 4 excellent
**Rating:** 7
**Confidence:** 3

**Summary:**

This paper proposes a decompositional approach to evaluate and improve a pre-trained text-to-image diffusion model given complex input prompts describing multiple objects and novel combinations. To that end, a LLM decomposes a difficult caption into disjoint assertions that can be individually evaluated using a VQA model. The assertion scores can be used to a) evaluate fine-grained image-text alignment and b) provide feedback for iterative refinement of generated images. The proposed metric correlates stronger with human ratings and the iterative refinement procedure increases the fidelity of generated images on complex input prompts.

**Strengths:**

- The paper is very well written and the method tackles an important problem
- The proposed DA-score has clear motivation and correlates much better with human ratings
- Using a LLM to produce a diverse pool of input prompts across varying complexity in terms of participating objects and realism is novel
- Quantitative and visual results are strong. The resulting model produces images that align a lot better on unreal/creative input captions.

**Weaknesses:**

I don't have any substantial weaknesses but including early approaches to combining text-to-image with VQA models would enhance the comprehensiveness of the related work section.
- https://dl.acm.org/doi/abs/10.1145/3372278.3390684
- https://aclanthology.org/2020.lantern-1.2/

**Questions:**

- TIFA is a recent, similar looking approach that uses VQA for evaluation. How does it differ?

**Limitations:**

- Yes, authors have extensively discussed limitations of the proposed method in the supplementary material.

---

> ### Author Rebuttal · Authors · 2023-08-09
>
> We thank the reviewer for their positive feedback and are pleased that they find our work well-written, novel, with clear motivation and having strong quantitative and visual results.
>
>
>
> >  I don’t have substantial weaknesses but including early approaches to combining text-to-image with VQA models would enhance the comprehensiveness of the related work section.
>
>
>
>  Thanks! We will update the related work section to include the suggested references in Sec. 2.
>
>
>
> >TIFA is a recent, similar looking approach that uses VQA for evaluation. How does it differ?
>
>
>
> **Please note that as also mentioned by Reviewer bVCd, TIFA is a recent arXiv preprint which is concurrent to our work.**
>
>
>
> Additionally, we note the following key differences between TIFA and the proposed approach:
>
> 1. In contrast with the proposed approach, TIFA mainly uses the idea of VQA for evaluation alone. The proposed approach further demonstrates how the interpretability of theses VQA scores can then be used to reliably improve the performance of text-to-image generation models.
>
> 2. TIFA directly uses the LLM model to generate questions with options. In contrast, we first break down the caption into a set of **disjoint** assertions, before rephrasing them into questions. In our experiments, we found the above approach to lead to less overlapping and redundant questions, which is especially critical for the use of DA-Score in the iterative refinement process.
>
>
>
> For instance, consider the following examples of question-generation outputs by TIFA and our approach:
>
>
>
> **Input Prompt1: "a woman making a documentary in a coral reef with a camera and a school of fish"**
>
> TIFA Questions:
>
> 1. is there a woman?
> 2. who is making a documentary?
> 3. is the woman making a documentary?
> 4. what is the woman doing?
> 5. is there a coral reef?
> 6. is there a camera?
> 7. what is the woman using?
> 8. is there a school of fish?
> 9. what is in the water?
>
> *Total LLM Tokens: 3681*
>
>
>
>
> DA-Score Questions:
>
> 1.  is there a woman in the image?
> 2.  is the woman making a documentary?
> 3.  is the image set in a coral reef?
> 4.  does the woman have a camera?
> 5.  is there a school of fish in the image?
>
> *Total LLM Tokens : 607*
>
>
>
> **Input Prompt2: "a crab using a sewing machine"**
>
> TIFA Questions:
>
> 1. is this a crab?
> 2. what animal is using the sewing machine?
> 3. is there a sewing machine?
> 4. what type of machine is this?
> 5. is the crab using the sewing machine?
>
> *Total LLM Tokens: 3563*
>
>
>
> DA-Score Questions:
>
> 1. is there a crab in the image?
> 2. is there a sewing machine in the image?
> 3. is the crab using the sewing machine?
>
> *Total LLM Tokens: 546*
>
>
>
> As seen above, we find that TIFA typically has multiple questions pertaining to the same concept (e.g., woman in example-1, crab in example-2). It also uses a secondary question filtering model to filter through the initial list of questions. This increases the number of LLM tokens required during inference (~4-6x compared with DAScore) and also incurs additional runtime cost. In contrast, we first break down an input prompt into a set of **disjoint** assertions and then rephrase them as questions. This allows DA-Score to generate a more targeted set of questions with a faster runtime which is highly critical for using DA-Score for iterative refinement with T2I models.

---

### Official Review · Reviewer_bVCd · 2023-07-04

**Soundness:** 3 good
**Presentation:** 3 good
**Contribution:** 3 good
**Rating:** 7
**Confidence:** 4

**Summary:**

This paper makes a twofold contribution: namely introducing a new metric for evaluating text-to-image alignment, as well as a method that builds upon this insight to improve the process of text-to-image generation.
For the first contribution of evaluating text-to-image alignment, the paper employs the strategy of using a Large Language Model (LLM) to decompose  the textual prompt into multiple components, and then using a Visual Question Answering (VQA) model to answer the individual components/assertions to obtain a final score for the alignment between the generated image and the textual prompt.
To improve the image generation process, the paper builds upon the existing approach Attend-and-Excite, that uses the cross-attention maps to focus on different tokens in the textual prompt. Here, the weights applied to the sub-prompts, as well as the strength of the cross-attention are used as parameters to be optimized in an iterative refinement procedure (while optimizing the decompositional metric) to obtain more faithful generated images.

**Strengths:**

1) The paper tackles an important problem (of both evaluating text-image alignment, and improving text-to-image generation), and proposed interesting ideas for the same.
2) The Prompt Decompositional Model intuitively makes a lot of sense, and quantitative results indicate that it seems to be significantly better than prior methods (i.e CLIPScore)
3) Using the improved evaluation metric to improve the generation is a good idea. While in terms of novelty, it is almost entirely based off Attend-and-Excite and prior work that uses prompt weighting, optimizing it during test-time is new, and seems to bring good improvements.
4) The experiments seems to be reasonably thorough. There are user studies that validate the key claims (of improved text-image alignment evaluation, and improved generation), as well as a new dataset of prompts.

**Weaknesses:**

One major concern I have with respect to the evaluation metric proposed in the paper is its similarity to [a]. While [a] is only an Arxiv preprint, and can definitely be considered as contemporary work, I would think that it is a good idea to atleast acknowledge it, since the idea behind breaking down a prompt into multiple Question-Answer pairs and then using a VQA model to evaluate it has been first introduced there.

In terms of the quantitative results, my worries are that the paper mostly relies on user studies/human evaluations to compare the proposed method against existing methods. While this is understandable (and quantitative results on the DAScore metric would be unfair, since the model is explicitly optimizing it), it might make sense to have quantitative results on other metrics (such as T2TScore) to demonstrate that the improvements are visible on automatic quantitative metrics. This would also allow benchmarking on other datasets beyond the Decomposable Prompts.

[a] Hu et al. TIFA: Accurate and Interpretable Text-to-Image Faithfulness Evaluation with Question Answering

**Questions:**

The only question I have is that all the experiments in this work are done on the Decomposable Prompts dataset. While it's large enough (with 4k prompts), I wonder if it is sufficient in terms of variety, of if it makes sense to also benchmark on other datasets such as DrawBench.

**Limitations:**

Yes, the limitations section in the supplementary material provides a fairly comprehensive overview of the possible limitations of the proposed method in the paper.

---

> ### Author Rebuttal · Authors · 2023-08-09
>
> We thank the reviewer for their positive feedback and are pleased that they find our work interesting, intuitive, tackling an important problem and with thorough experiments.
>
>
>
> >  While [a] is only an Arxiv preprint, and can definitely be considered as contemporary work, I would think that it is a good idea to acknowledge it.
> >
> >  [a] Hu et al. TIFA: Accurate and Interpretable Text-to-Image Faithfulness Evaluation with Question Answering
>
>
>
>  Thanks! We will update Sec. 2 (Related Work) to include the same as *contemporary* work.
>
> Please also note that [a] only uses the idea of VQA for evaluation alone. The proposed approach further demonstrates how the interpretability of computed VQA scores can then be used to reliably improve the performance of text-to-image generation models.
>
>
>
> >  While this is understandable (and quantitative results on the DAScore metric would be unfair, since the model is explicitly optimizing it), it might make sense to have quantitative results on other metrics (such as T2TScore) to demonstrate that the improvements are visible on automatic quantitative metrics.
>
>
>
> We predominantly use human-annotations/user-studies to avoid erroneous evaluation due to the weak-correlation between prior image-text alignment metrics and human preferences (Fig. 4, main paper).  Nevertheless, as seen in Table below, we do find that increases in DA-Score are also accompanied by increase in other evaluation metrics such as T2T Score during the iterative refinement process.
>
>
>
> | Method                   | T2T-Score |
> | ------------------------ | :-------- |
> | Stable-Diffusion [1]     | 72.11     |
> | Composable-Diffusion [6] | 71.19     |
> | Structure-Diffusion [8]  | 72.84     |
> | Attend-and-Excite [7]    | 74.23     |
> | Ours (PW + CA)           | 76.09     |
>
>
>
>
>
> >  While Decomposable-Captions dataset is large enough (with 4k prompts), I wonder if it is sufficient in terms of variety, or if it makes sense to also benchmark on other datasets such as DrawBench.
>
>
>
> **Diversity and challenging nature of Decomposable Captions Dataset**
>
> Please note that as mentioned in Sec. 3 (main paper) and dataset details in App. C (Fig. 9, 10, 11 from SM), the decomposable-captions-4k dataset consists of prompts across a diverse range of complexities in both **1)** number of subjects, and, **2)** the realism of the input prompts. This allows us to evaluate the efficacy of our approach over a much wider data distribution as compared to prior works [e.g., Attend-and-Excite], which predominantly analyse the performance on relatively simple prompts with two subjects (e.g. object a and object b).
>
>
>
> The diversity and challenging nature of the Decomposable-Captions dataset is also revealed by variation of the overall alignment performance on different subsets of the introduced dataset. For instance, while the alignment accuracy for Stable-Diffusion on the *easy* realism subset is as high as 73.4%,  as the *realism* difficulty increases the alignment accuracy drops down to 21.6%. This value further drops down to 9.94% when considering only prompts with *number of subjects = 5* and *realism difficulty = 'very hard'*.
>
>
>
> The above variation (along with dataset details in App. C - Fig. 9, 10, 11), not only point to the diversity of the overall dataset but also provide an idea of the challenging nature of the introduced dataset.
>
>
>
> **Additional evaluation on DrawBench**
>
> In addition to above, we also perform additional experiments evaluating the proposed approach on Drawbench prompts.
>
>
>
> **Results for DA-Score**:  We use the Stable Diffusion v1.5 model to generate images corresponding to the prompts from the DrawBench dataset (excluding rare words and misspellings as the same are not usable with VQA). Each prompt-image pair is then annotated by human experts on a scale of 1-5 (5: image is an accurate match for the prompt). The correlation results across different evaluation metrics are shown below. We observe that similar to decomposable-captions dataset, the DA-Score shows significantly higher correlation with human preferences as compared to prior evaluation metrics.
>
> | Evaluation Method              | Spearman's $\rho$ (%) | Kendall's $\tau$ (%) |
> | ------------------------------ | --------------------- | -------------------- |
> | BLIP-Score [10]                | 19.02                 | 14.07                |
> | BLIP2-Score [11]               | 21.49                 | 16.20                |
> | BLIP-R ITM Score [10]          | 20.14                 | 14.90                |
> | BLIP2-R ITM Score [11]         | 22.64                 | 16.99                |
> | T2T-Score [7]                  | 24.33                 | 18.13                |
> | DA-Score (with BLIP-VQA) | 55.01                 | 43.21                |
>
>
>
> **Results for Iterative Refinement**:
>
> While fullscale quantitative evaluation with human evaluation across all methods was not feasible during the response period, we provide detailed qualitative comparisons with prompts from the Drawbench dataset in Fig. 3 (global response pdf).  Similar to observations on the Decomposable-Captions-4k dataset, we find that the use of iterative refinement helps improve the text-to-image alignment across a diverse range of input prompts.
>
>
>
> Please also note that while the proposed approach performs better than prior work, the improvements noticed with iterative refinement are dependent on the ability of the VQA model to detect such inconsistencies. Thus, prompts with inconsistencies not detectable by a VQA model e.g., misspellings, rare words etc, do not observe significant/noticeable improvement through the use of iterative refinement.

---

> > ### Comment · Reviewer_bVCd · 2023-08-18
> >
> > I thank the authors for their detailed response, addressing all the remaining concerns. I have raised my score to 7 on account of this.

---

### Official Review · Reviewer_sTZE · 2023-07-06

**Soundness:** 3 good
**Presentation:** 3 good
**Contribution:** 3 good
**Rating:** 5
**Confidence:** 4

**Summary:**

The paper proposes a method called Decompositional-Alignment-Score (DA-Score) to assess and enhance the alignment between text and generated images during training. The authors accomplish this by breaking down each prompt or text into a series of assertions using a large language model (LLM). They then calculate correlation scores between each assertion and the generated image, employing a pre-trained VQA model. The final score, obtained by combining all the assertion scores, serves as an evaluation metric for text-image alignment. It further guides the generative models to produce more aligned images iteratively.

**Strengths:**

This paper seeks to address a significant challenge in the text-to-image generation task, i.e., text-image alignment. The proposed framework adopts LLM and pre-trained multi-modal models, which is reasonable.

**Weaknesses:**

1. There is a concern regarding the performance of the VQA model, as it plays a crucial role in determining the accuracy of the proposed score. The effectiveness and reliability of the evaluation process heavily rely on the capabilities and performance of the VQA model.
2. The experiments are not convincing enough and there and there is a lack of some specific details regarding the design of the dataset.

Please refer to more details in the following.


**Questions:**

*Method
1. The performance of the proposed alignment score heavily relies on the accuracy of the VQA's answering results (also mentioned in the limitations by the authors). It raises the question of whether the used VQA model is sufficiently robust to handle all the given questions. If available, it would be helpful to provide quantitative results or an assessment of the accuracy of the VQA model in addressing the given questions on the dataset in this paper.
2. It seems there is a lack of definition for the meaning of the variable $l$ in Eq.(5).

*Experiment and Dataset
1. How about the quality of the generated images? Based on the visual results presented in the paper (e.g., Figure 5), it appears that focusing solely on text-image alignment may lead to a reduction in image quality. It would be beneficial to provide additional results that consider image quality evaluation metrics such as FID and FID-CLIP.
2. It seems that the generated images of the baseline generative models (e.g., stable diffusion) are not satisfied enough. I think it would be more realistic if use a more recent version like stable diffusion 2.1. Besides, it raises the question of whether the proposed alignment method remains effective or necessary when applied to more powerful text-to-image generation models, such as Imagen [1] or Parti [2].
3. The paper does not provide clarity on the number of annotators involved in rating the samples within the proposed dataset. If only a small number of annotators, the obtained human annotations may lack sufficient credibility and be susceptible to subjective biases.

[1] Saharia, Chitwan, et al. Photorealistic text-to-image diffusion models with deep language understanding. Advances in Neural Information Processing Systems 35 (2022): 36479-36494.

[2] Yu, Jiahui, et al. "Scaling autoregressive models for content-rich text-to-image generation." arXiv preprint arXiv:2206.10789 (2022).


**Limitations:**

The authors discuss three limitations of the current version and provide the corresponding proposal for a solution.

---

> ### Author Rebuttal · Authors · 2023-08-10
>
> >   If available, it would be helpful to provide quantitative results or an assessment of the accuracy of the VQA model in addressing the given questions on the dataset in this paper.
>
>  To analyse the accuracy of the VQA model's answering results, we perform an additional user-study wherein for each question-image pair, the human subject is asked to answer as yes, no or maybe (when answer is unclear). The accuracy of different VQA models is then predicted using the human labels as ground truths. Results are shown below.
>
> | VQA Model      | Accuracy % | Avg. Inference Time with DA-Score (s) |
> | -------------- | ---------- | ------------------------------------- |
> | VILT-VQA [13]  | 76.8%      | 0.224 s                               |
> | BLIP-VQA [10]  | 81.2%      | 0.329 s                               |
> | BLIP2-VQA [11] | 83.4%      | 0.512 s                               |
>
> We observe that while using a better VQA model (e.g. BLIP-2) obtains a better answering accuracy, using a relatively lightweight BLIP-VQA model helps us reduce the overall inference time during evaluation while still maintaining competitive performance on the human-correlation scores (Fig. 4, main paper).
>
> >  It seems there is a lack of definition for the meaning of the variable $l$ in Eq.(5).
>
> Apologies for the typo. As described in Alg. 2 (SM), Eq. 5 should be as follows (to increase the weight of assertion with lowest alignment score):
>
> $$w_i^{k+1} =
>     \begin{cases}
>     w_i^k + \Delta, \ \text{if} \quad  i = \text{argmin}_{l}  \ u_l(\mathcal{I},\mathcal{P})\\\\
>     w_i^k \quad \text{otherwise}
>     \end{cases},$$
>
> >  It would be beneficial to provide additional results that consider image quality evaluation metrics.
>
> Since the ground truth real images (required for FID) corresponding to typically creative and challenging prompts (e.g. a lion playing a piano) on the decomposable-captions dataset are not available, we use the following results to evaluate overall image quality:
>
> 1. Quantitative Results: Similar to [8] we perform an additional user study for image fidelity, wherein given a pair of images (ours vs prior work), the human subject is asked to select the image with higher visual quality. The user-study was conducted across 1000 input-prompts randomly sampled from decomposable-captions dataset. Results are reported in Table below.
>
> 2. Qualitative Results: We also provide additional qualitative results comparing the output image quality across different methods. Results are shown in Fig. 1 (global response).
>
>
> | Method                   | Win (Ours is better)   | Tie  | Loss (Ours is worse) |
> | ------------------------ | :--: | :--: | :--: |
> | Stable-Diffusion [1]     | 6.9  | 78.4 | 14.7 |
> | Composable-Diffusion [6] | 38.8 | 52.9 | 8.3  |
> | Attend-and-Excite [7]    | 23.6 | 69.7 | 6.7  |
>
> As shown in Table above and Fig. 1 (global response), we observe that while the proposed iterative refinement approach does lead to a slight reduction in image quality when compared to Stable-Diffusion (6.9 % win, 78.4 % tie, 14.7 % lose), it still shows better visual quality than prior works like Attend-and-Excite and Composable-Diffusion.
>
> We find that the reduction in visual quality typically occurs when performing large number of modifications (e.g. cross-attention updates) to the reverse diffusion process. In contrast with [6,7], our approach is able to adaptively adjust the number of such modifications by monitoring the proposed DA-Score. This helps us achieve better alignment results while still maintaining superior image quality as compared with prior works.
>
>
> >  It seems that the generated images of the baseline generative models (e.g., stable diffusion) are not satisfied enough. I think it would be more realistic if use a more recent version like stable diffusion 2.1.
>
> Please note that in order to maintain consistency (implementation hyperparams) with prior works [Attend-Excite, Structure-Diffusion], we use the open-source Stable-Diffusion v1.4 model for our experiments.
>
> Nevertheless, we note that the proposed approach is model-agnostic and easily applicable to other diffusion models. For instance, results with use of proposed iterative refinement on more recent Stable Diffusion models (v1.5 and v2.1) are shown in Fig. 2  (global response). We observe that while more recent versions of Stable-Diffusion show better performance, the generated images often have inconsistencies (e.g., pool and beach chair in example-1, bowtie and surfboard in example-2) with the input prompt. The proposed approach is able to detect such inconsistencies (with DA-Score) and gradually improve the final image output through iterative refinement.
>
> >  The paper does not provide clarity on the number of annotators involved in rating the samples within the proposed dataset.
>
> In order to ensure the quality of collected fine-grain human ratings (scale: 1-5 between synthetic images and complex prompts with multiple subjects), the data annotations were collected from a group of 20 *"expert"* annotators (similar to contemporary work [a]) with experience in image-text rating tasks. An unlimited time is provided for each annotation.
>
> We also verify the quality of the collected finegrain annotations by performing a secondary user-study for the simpler binary task of evaluating alignment accuracy (i.e. whether an image accurately matches the text prompt?) on a larger pool of 100 annotators. Results are shown below. We observe that the results collected among the larger group (with binary task of evaluating alignment accuracy) shows similar results to those obtained from the fine-grain expert human ratings (rating=5).
>
> |Method|Alignment Acc. [Expert Rating=5]|Alignment Acc. [Binary Task with 100 Subjects]|
> |---|---|---|
> |Stable-Diffusion [1] |43.66|45.73|
> |Attend-and-Excite [7] |65.50|65.05|
> |Ours (PW+CA)|74.16|73.24|
>
> [a] TIFA: Accurate and Interpretable Text-to-Image Faithfulness Evaluation with Question Answering, Hu et al., 2023.

---

### Official Review · Reviewer_GK7B · 2023-07-07

**Soundness:** 3 good
**Presentation:** 4 excellent
**Contribution:** 3 good
**Rating:** 7
**Confidence:** 4

**Summary:**

The paper proposes a new approach to evaluate text-image alignment of generated images and how to use that feedback to improve the image quality. For this, a LLM is first used to parse the text description into individual assertions and the alignment of those individual assertions with the generated image is then measured via a VQA model. The resulting metric is shown to better correlate with human judgement than other metrics such as CLIP/BLIP score. Furthermore, the feedback of the VQA model can be used to increase the impact of specific parts of the promp during the denoising process which improves the text-image alignment of the generated image without requiring additional model training.

**Strengths:**

The paper is well written and presented. The novel evaluation process is intuitive and automated, and shows better alignment with human judgement than other metrics. The use of the metric to improve tet-image alignment at test time is also shown and leads to better results than other approaches. The evaluation includes several human user studies highlighting the merits of the new approach.

**Weaknesses:**

The approach introduces the need for two additional networks (LLM and VQA model) into the pipeline which will incur additional cost and compute requirements. The overall quality and the resulting scores are likely also heavily dependent on the quality of the LLM and VQA model.

**Questions:**

While the approach works I wonder how much slower it is (as an evaluation metric) compared to other approaches such as CLIP. CLIP only requires a forward pass through the text and image encoder, whereas the new approach requires interactions with an LLM (to get the assertions) and then also needs to query a VQA model for each assertion.

**Limitations:**

Overall this is a good paper and the evaluation metric reminds me of an updated and improved version of the SOA score (Semantic object accuracy for generative text-to-image synthesis, TPAMI, 2020).

---

> ### Author Rebuttal · Authors · 2023-08-09
>
> We thank the reviewer for their positive feedback and are pleased that they find our work well written, intuitive and comprehensive with several user studies.
>
>
>
> >  The overall quality and the resulting scores are likely also heavily dependent on the quality of the LLM and VQA model.
>
>  Yes, as discussed as limitation in the supplementary material,  we do note that the performance of our approach depends on assertion-parsing from the LLM and question-answering from the VQA model. Nevertheless, as shown through the quantitative results,   competitive results across a diverse range of input prompts are obtained in our current framework by using BLIP model for VQA and GPT-3.5-Turbo for text-prompt parsing.
>
>
>
> Furthermore, as noted in SM and response to Q1@sTZE, we find that while the use of a more stronger VQA model (e.g., BLIP2-VQA) can help improve overall question-answering performance, the use of relatively lightweight BLIP-VQA model (used in the paper) still allows us to achieve competitive performance without sacrificing runtime performance during the iterative refinement process.
>
>
>
> >   While the approach works I wonder how much slower it is (as an evaluation metric) compared to other approaches such as CLIP.
>
>  Please see Table below for a detailed comparison of the average inference times when evaluating image-text alignment scores across a random sample of 1000 image-prompt pairs from the Decomposable-Captions-4k dataset.
>
> Since the overall inference time would also depend on the nature of LLM used and the API speed, average inference times with DA-Score are reported in 2 forms 1) DA-Score (with precomputed assertions): which reportes the average inference time when using precomputed assertions, and 2) DA-Score (Overall): where both assertions and VQA based scores are computed at inference time. We use parallel API requests to the "gpt-3.5-turbo" model with 64 processes to reduce the average overhead required for obtaining assertions from the LLM model.
>
> | Evaluation Method            | Avg. Inference Time (s) |
> | ---------------------------- | ----------------------- |
> | CLIP-Score [9]               | 0.061 sec               |
> | BLIP-Score [10]              | 0.079 sec               |
> | BLIP2-Score [11]             | 0.112 sec               |
> | BLIP-R ITM Score [10]        | 0.082 sec               |
> | BLIP2-R ITM Score [11]       | 0.116 sec               |
> | T2T-Score [7]                | 0.438 sec               |
> | DA-Score (Precomputed Ques.) | 0.329 sec               |
> | DA-Score (Overall)           | 0.457 sec               |
>
>
>
> We observe that:
>
> 1. While the proposed evaluation metric (DA-Score) is slower when compared to single model metrics (e.g. CLIP, BLIP, BLIP2), DA-Score with precomputed assertions is faster than T2T-Score (which first generates multiple captions for the generated image using BLIP-captioning model and then computes an average similarity score using CLIP), while exhibiting significantly higher correlation with human preference ratings (please refer Fig. 4 main paper).
>
> 2. Also in context of iterative refinement, we note that the question generation needs to be done only once per text-prompt. Furthermore, the average time required for DA-Score computation is significantly lower than the runtime for a standard reverse diffusion process. This allows DA-Score to serve as a quick evaluation metric when performing iterative refinement, thereby retaining its usability for practical applications.

---

### Author Rebuttal · Authors · 2023-08-10

We thank all the reviewers for their positive feedback and are pleased that they find our work well written [GK7B, 5iW3], intuitive [GK7B, bVCd],  tackling an important problem [sTZE, bVCd, 5iW3] and comprehensive with several user studies [GK7B,bVCd] and strong quantitative, visual results [5iW3].



We have tried our best to answer all the queries raised during the rebuttal through individual responses to each reviewer. Additionally, we also provide a global response pdf containing the requested results as summarised below:

* Additional image quality comparison results [sTZE]: Fig. 1
* Generalizability to more recent Stable-Diffusion versions [sTZE]: Fig. 2
* Additional results on DrawBench prompts [bVCd]: Fig. 3


We hope that the additional results / clarifications answer the questions raised during the discussion period.

---

> ### Comment · Area_Chair_dMhB · 2023-08-21
>
> Thank you for this comment and elaborating the required results in the rebuttal as well. We will take it into consideration while making the final decision regarding the paper.

---

### Decision · Program_Chairs · 2023-09-21

**Decision:**

Accept (poster)

**Comment:**

This paper focuses on an important and challenging problem - the evaluation of text-to-image generation. All reviewers are positive about this submission. Three voted for Accept and one voted for a Borderline Accept. The strengths of this paper are clear and confirmed by all the reviewers, including well-written and presented, novel techniques and solid experiments. The authors also did a good rebuttal to solve reviewers' concerns on speed and similarity to some existing works. These valuable responses and discussions should be included in the final camera-ready version. Based on the reviewer's comments and discussion, the AC voted for acceptance of this submission.